# Optimal Control Via Neural Networks: A Convex Approach

**Yize Chen,**[*] **Yuanyuan Shi,**[*] **Baosen Zhang**
Department of Electrical and Computer Engineering,
University of Washington,
Seattle, WA 98195, USA
`{yizechen, yyshi, zhangbao}@uw.edu`

## Abstract

Control of complex systems involves both system identification and controller design. Deep neural networks have proven to be successful in many identification tasks, however, from model-based control perspective, these networks are difficult to work with because they are typically nonlinear and nonconvex. Therefore many systems are still identified and controlled based on simple linear models despite their poor representation capability. In this paper we bridge the gap between model accuracy and control tractability faced by neural networks, by explicitly constructing networks that are convex with respect to their inputs. We show that these input convex networks can be trained to obtain accurate models of complex physical systems. In particular, we design input convex recurrent neural networks to capture temporal behavior of dynamical systems. Then optimal controllers can be achieved via solving a convex model predictive control problem. Experiment results demonstrate the good potential of the proposed input convex neural network based approach in a variety of control applications. In particular we show that in the MuJoCo locomotion tasks, we could achieve over 10% higher performance using $5\times$ less time compared with state-of-the-art model-based reinforcement learning method; and in the building HVAC control example, our method achieved up to 20% energy reduction compared with classic linear models.

## 1 Introduction

Decisions on how to best operate and control complex physical systems such as the power grid, commercial and industrial buildings, transportation networks and robotic systems are of critical societal importance. These systems are often challenging to control because they tend to have complicated and poorly understood dynamics, sometimes with legacy components are built over a long period of time (Wolf, 2009). Therefore detailed models for these systems may not be available or may be intractable to construct. For instance, since buildings account for 40% of the global energy consumption (Cheng et al., 2008), many approaches have been proposed to operate buildings more efficiently by controlling their heating, ventilation, and air conditioning (HVAC) systems (Zhang et al., 2017). Most of these methods, however, suffer from two drawbacks. On one hand, a detailed physics model of a building can be used to accurately describe its behavior, but this model can take years to develop. On the other hand, simple control algorithms have been developed by using linear (RC circuit) models (Ma et al., 2012) to represent buildings, but the performance of these models may be poor since the building dynamics can be far from linear (Shaikh et al., 2014).

In this paper, we leverage the availability of data to strike a balance between requiring painstaking manual construction of physics based models and the risk of not capturing rich and complex system dynamics through models that are too simplistic. In recent years—with the growing deployment of sensors in physical and robotics systems—large amount of operational data have been collected, such as in smart buildings (Suryadevara et al., 2015), legged robotics (Meger et al., 2015) and manipulators (Deisenroth et al., 2011). Using these data, the system dynamics can be learned directly and then automatically updated at periodic intervals. One popular method is to parameterize these

---

[*]Authors contribute equally.

complex system dynamics using deep neural networks to capturing complex relationships (He et al., 2016; Vaswani et al., 2017), yet few research investigated how to integrate deep learning models into real-time closed-loop control of physical systems.

A key reason that deep neural networks have not been directly applied in control is that even though they provide good performances in learning system behaviors, optimization on top of these networks is challenging (Kawaguchi, 2016). Neural networks, because of their structures, are generally not convex from input to output. Therefore, many control applications (e.g., where real-time decisions need to be made) choose to favor the computational tractability offered by linear models despite their poor fitting performances.

In this paper we tackle the modeling accuracy and control tractability tradeoff by building on the input convex neural networks (ICNN) in (Amos et al., 2017) to both represent system dynamics and to find optimal control policies. By making the neural network convex from input to output, we are able to obtain *both good predictive accuracies and tractable computational optimization problems*. The overall methodology is shown in Fig. 1. Our proposed method (shown in Fig. 1 (b)) firstly utilizes an input convex network model to learn the system dynamics and then computes the best control decisions via solving a convex model predictive control (MPC) problem, which is tractable and has optimality guarantees. This is different from existing methods that uses model-free end-to-end controller which directly maps input to output (shown in Fig. 1 (a)). Another major contribution of our work is that we explicitly prove that ICNN can represent all convex functions and systems dynamics, and is *exponentially* more efficient than widely used convex piecewise linear approximations (Magnani & Boyd, 2009).

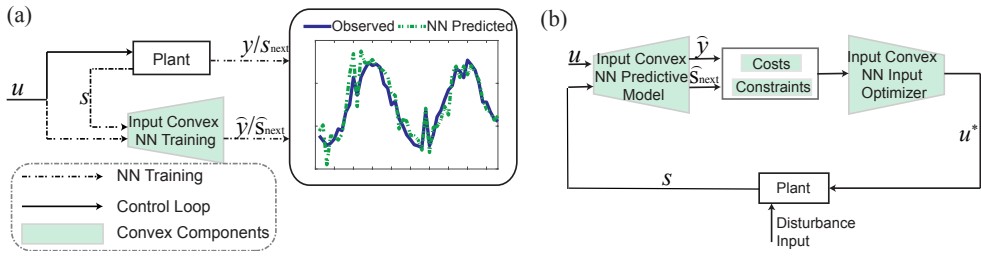

Figure 1: Our proposed model-based method, (a) an input convex neural network is first trained to learn the system dynamics, then (b) we solve a convex predictive control problem to find the optimal actions which are input convex neural networks' inputs. The optimization steps are also based on objectives and dynamics constraints represented by the trained networks.

## 1.1 RELATED WORK

The work in (Amos et al., 2017) was an impetus for this paper. The key differences are that the goal in (Amos et al., 2017) is to show that ICNN can achieve similar classification performances as conventional neural networks and how the former can be used in inference and prediction problems. Our goal is to use these networks for optimization and closed-loop control, and in a sense that we are more interested in the overall system performances and not directly the performance of the networks. We also extend the class of networks to include RNNs to capture dynamical systems.

Control and decision-making have used deep learning mainly in model-free end-to-end controller settings (shown in Fig. 1 (a)), such as sequential decision making in game (Mnih et al., 2013), robotics manipulation (Levine & Koltun, 2014; Levine et al., 2016), and control of cyber-physical systems (Wei et al., 2017; O'Neill et al., 2010). However, much of the success relies heavily on a reinforcement learning setup where the optimal state-action relationship can be learned via a large number of samples. However, many physical systems do not fit into the reinforcement learning process, where both the sample collection is limited by real-time operations, and there are physical model constraints hard to represent efficiently.

To address the above sample efficiency, safety and model constraints incompatibility concerns faced by model-free reinforcement learning algorithms in physical system control, we consider a model-based control approach in this work. Model-based control algorithms often involve two stages – system identification and controller design. For the system identification stage, the goal

is to learn a fixed form of system model to minimize some prediction error (Ljung, 1998). Most efficient model-based control algorithms have used a relatively simple function estimator for the system dynamics identification (Nagabandi et al., 2018), such as linear model (Ma et al., 2012) and Gaussian processes (Meger et al., 2015; Deisenroth et al., 2011). These simplified models are sample-efficient to learn, and can be nicely incorporated in the sub-sequent optimal control problems. However, such simple models may not have enough representation capacity in modeling large-scale or high-dimension systems with nonlinear dynamics. Deep neural networks (DNNs) feature powerful representation capability, while the main challenge of using DNNs for system identification is that such models are typically highly non-linear and non-convex (Kawaguchi, 2016), which causes great difficulty for following decision making. A recent work from (Nagabandi et al., 2018) is close in spirit as our proposed method. Similarly, the authors use a model-based approach for robotics control, where they first fit a neural network for the system dynamics and then use the fitted network in an MPC loop. However, since (Nagabandi et al., 2018) use conventional NN for system identification, they cannot solve the MPC problem to global optimality. Our work shows how the proposed ICNN control algorithm achieves the benefits from both sides of the world. The optimization with respect to inputs can be implemented using off-the-shelf deep learning optimizers, while we are able to obtain good identification accuracies and tractable computational optimization problems by using proposed method at the same time.

## 2 CLOSED-LOOP CONTROL WITH INPUT CONVEX NEURAL NETWORKS

In this paper, we consider the settings where a neural network is used in a closed-loop system. The fundamental goal is to optimize system performance which is beyond the learning performance of network on its own. In this section we describe how input convex neural networks (ICNN) can be extremely useful in these systems by considering two related problems. First, we show how ICNN perform in single-shot optimization problems. Then we extend the results to an input convex *recurrent* neural networks (ICRNN), which allows us to both capture systems' complex dynamics and make time-series decisions.

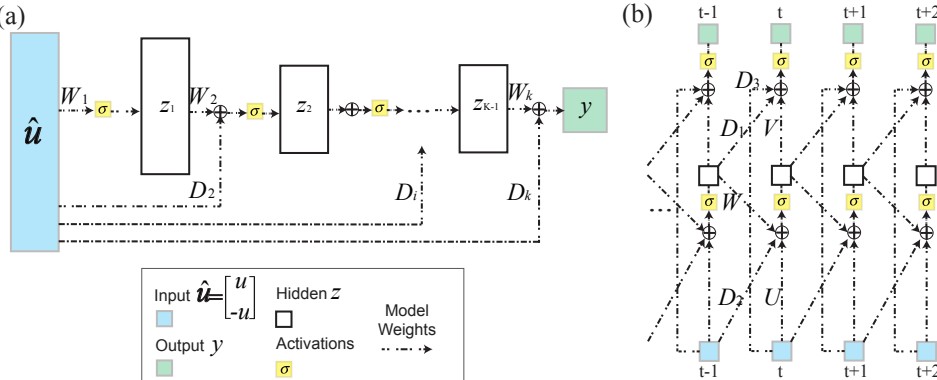

Figure 2: Input convex neural network. Input convex neural network. (a) Input convex feed-forward neural networks (ICNN). One notable addition is the direct "passthrough" layers $\mathbf{D}_{2:k}$ that connect the inputs to hidden units for better model representation ability. (b) The proposed input convex recurrent neural networks (ICRNN) architectures. In our control settings, we keep all weights in both networks nonnegative, while expanding the inputs with $-\mathbf{u}$.

### 2.1 SINGLE-SHOT PROBLEM

The following proposition states a simple sufficient condition for a neural network to be input convex:

**Proposition 1.** *The feedforward neural network in Fig. 2(a) is convex from input to output given that all weights between layers $\mathbf{W}_{1:k}$ and weights in the "passthrough" layers $\mathbf{D}_{2:k}$ are non-negative, and all of the activation functions are convex and nondecreasing (e.g. ReLU).*

The structure of the input convex neural network (ICNN) structure in Proposition 1 is motivated by the structure in (Amos et al., 2017) but modified to be more suitable to control of dynamical systems. In (Amos et al., 2017) it only requires $\mathbf{W}_{2:k}$ to be non-negative while having no restrictions on weights $\mathbf{W}_1$ and $\mathbf{D}_{2:k}$. Our construction achieves the exact representation by expanding the inputs to include both $\mathbf{u}$ ($\in \mathbb{R}^d$) and $-\mathbf{u}$. Then any negative weights in $\mathbf{W}_1$ and $\mathbf{D}_{2:k}$ in (Amos et al., 2017)'s ICNN structure is set to zero and its negation (which is positive) is added as the weight for corresponding $-\mathbf{u}$. The reason for our construction is to allow the network to be "rolled out in time" when we are dealing with dynamical systems and multiple networks need to be composed together.

An simple example that demonstrates how the proposed ICNN can be used to fit a convex function comes form fitting the $|u|$ function. This function is convex and both decreasing and increasing. Let the activation function be $ReLU(\cdot) = \max(\cdot, 0)$. We can write $|u| = -u + 2ReLU(u)$ (Amos et al., 2017). However, in this representation, we need a negative weight, the $-1$ in front of $u$, and this would be troublesome if we compose several networks together. In our proposed ICNN structure with all positive weights and input negation duplicates, we can write $|u| = v + 2ReLU(u)$, where we impose a constraint $v = -u$. Such doubline on the number of input variables may potentially make the network harder to train. Yet during control, having all of the weights positive maintains the convexity between inputs and outputs even if multiple steps are considered which will be discussed in Section 2.2. The constraint $v = -u$ is linear and can be easily included in any convex optimization.

This proposition follows directly from composition of convex functions (Boyd & Vandenberghe, 2004). Although it allows for any increasing convex activation functions, in this paper we work with the popular ReLU activation function. Two notable additions in ICNN compared with conventional feedforward neural networks are: 1) Addition of the direct *"passthrough" layers* connecting inputs to hidden layers and conventional *feedforward layers* connecting hidden layers for better representation power. 2) the expanded inputs that include both $\mathbf{u}$ and $-\mathbf{u}$. The proposed ICNN structure is shown in Fig. 2(a). Note that such construction guarantees that the network is convex and non-decreasing with respect to the expanded inputs $\hat{\mathbf{u}} = \begin{bmatrix} \mathbf{u} \\ -\mathbf{u} \end{bmatrix}$, while the output can achieve either decreasing or non-decreasing functions over $\mathbf{u}$.

Fundamentally, ICNN allows us to use neural networks in decision making processes by guaranteeing the solution is unique and globally optimal. Since many complex input and output relationships can be learned through deep neural networks, it is natural to consider using the learned network in an optimization problem in the form of

$$\min_{\mathbf{u}} f(\mathbf{u}; \mathbf{W}) \tag{1a}$$

$$\text{s.t. } \mathbf{u} \in \mathscr{U}, \tag{1b}$$

where $\mathscr{U}$ is a convex feasible space. Then if $f$ is an ICNN, optimizing over $\mathbf{u}$ is a convex problem, which can be solved efficiently to global optimality. Note that we will always duplicate the variables by introducing $\mathbf{v} = -\mathbf{u}$, but again this does not change the convexity of the problem. Of course, since the weights of the network are restricted to be nonnegative, the performance of the network (e.g., classification) may be worse. A common thread we observe in this paper is that trading off classification performance with tractability can be preferable.

## 2.2 CLOSED-LOOP CONTROL AND RECURRENT NEURAL NETWORKS

In addition to the single-shot optimization problem in (1), we are interested in optimally controlling a dynamical system. To model the temporal dependency of the system dynamics, we propose to use recurrent neural networks (instead of feed-forward neural networks). Recurrent networks carry an internal state of the system, which introduces coupling with previous inputs to the system. Fig. 2(b) shows the proposed input convex recurrent neural networks (ICRNN) structure. This network maps from input $\hat{\mathbf{u}}$ to output $y$ with memory unit $\mathbf{z}$ according to the following Eq. (2),

$$\mathbf{z}_t = \sigma_1 (\mathbf{U}\hat{\mathbf{u}}_t + \mathbf{W}\mathbf{z}_{t-1} + \mathbf{D}_2\hat{\mathbf{u}}_{t-1}), \tag{2}$$

$$y_t = \sigma_2 (\mathbf{V}\mathbf{z}_t + \mathbf{D}_1\mathbf{z}_{t-1} + \mathbf{D}_3\hat{\mathbf{u}}_t), \tag{3}$$

where $\hat{\mathbf{u}} = \begin{bmatrix} \mathbf{u} \\ -\mathbf{u} \end{bmatrix}$, and $D_1, D_2, D_3$ are added direct "passthrough" layers for augmenting representation power. If we unroll the dynamics with respect to time, we have $y_t = f(\hat{\mathbf{u}}_1, \hat{\mathbf{u}}_2, ..., \hat{\mathbf{u}}_t; \theta)$ where $\theta =$

$[\mathbf{U}, \mathbf{V}, \mathbf{W}, \mathbf{D_1}, \mathbf{D_2}, \mathbf{D_3}]$ are network parameters, and $\sigma_1, \sigma_2$ denote the nonlinear activation functions. The next proposition states a sufficient condition for the network to be input convex.

**Proposition 2.** *The network shown in Fig. 2(b) is a convex function from inputs to output if all weights $U,V,W,D_1,D_2,D_3$ are non-negative, and all activation functions are convex and nondecreasing (e.g. ReLU).*

The proof of this proposition again follows directly from the composition rule of convex functions. Similarly to the ICNN case, by expanding the inputs vector to include both $\mathbf{u}$ and $-\mathbf{u}$ and restricting all weights to be non-negative, the resulted ICRNN structure is a convex and non-decreasing mapping from inputs to output.

The proposed ICRNN structure can be leveraged to represent system dynamics for close-loop control. Consider a physical system with discrete-time dynamics, at time step $t$, let's define $\mathbf{s}_t$ as the *system states*, $\mathbf{u}_t$ as the *control actions*, and $y_t$ as the *system output*. For example, for the real-time control of a building system, $\mathbf{s}_t$ includes the room temperature, humidity, etc; $\mathbf{u}_t$ denotes the building appliance scheduling, room temperature set-points, etc; and output $y_t$ is the building energy consumption. In addition, there maybe exogenous variables that impact the output of the system, for example, outside temperature will impact the energy consumption of the building. However, since the exogenous variables are not impacted by any of the control actions we take, we suppress them in the formulation below. The time evolution of a system is described by

$$y_t = f(\mathbf{s}_t, \mathbf{u}_t), \tag{4a}$$

$$\mathbf{s}_{t+1} = g(\mathbf{s}_t, \mathbf{u}_t) \tag{4b}$$

where (4b) describes the coupling between the current inputs to the future system states. Physical systems described by (4) may have significant inertia in the sense that the outcome of any control actions is delayed in time and there are significant couplings across time periods.

Since we use ICRNNs to represent both the system dynamics $g(\cdot)$ and the output $f(\cdot)$, the control variable $\mathbf{u}$ expands as $\hat{\mathbf{u}}$. The optimal receding horizon control problem at time $t$ can be written as,

$$\underset{\mathbf{u}_t, \mathbf{u}_{t+1}, \ldots, \mathbf{u}_{t+T}}{\text{minimize}} \quad C(\hat{\mathbf{x}}, \mathbf{y}) = \sum_{\tau=t}^{t+T} J(\hat{\mathbf{x}}_\tau, y_\tau) \tag{5a}$$

$$\text{subject to} \quad y_\tau = f(\hat{\mathbf{x}}_{\tau-n_w}, \hat{\mathbf{x}}_{\tau-n_w+1}, \ldots, \hat{\mathbf{x}}_\tau), \forall \tau \in [t, t+T] \tag{5b}$$

$$\mathbf{s}_\tau = g(\hat{\mathbf{x}}_{\tau-n_w}, \hat{\mathbf{x}}_{\tau-n_w+1}, \ldots, \hat{\mathbf{x}}_{\tau-1}, \hat{\mathbf{u}}_\tau), \forall \tau \in [t, t+T] \tag{5c}$$

$$\hat{\mathbf{x}}_\tau = \begin{bmatrix} \mathbf{s}_\tau \\ \hat{\mathbf{u}}_\tau \end{bmatrix}, \hat{\mathbf{u}}_\tau = \begin{bmatrix} \mathbf{u}_\tau \\ \mathbf{v}_\tau \end{bmatrix}, \forall \tau \in [t, t+T] \tag{5d}$$

$$\mathbf{v}_\tau = -\mathbf{u}_\tau, \forall \tau \in [t, t+T] \tag{5e}$$

$$\mathbf{s}_\tau \in \mathscr{S}_{feasible}, \forall \tau \in [t, t+T] \tag{5f}$$

$$\mathbf{u}_\tau \in \mathscr{U}_{feasible}, \forall \tau \in [t, t+T] \tag{5g}$$

where a new variable $\hat{\mathbf{x}} = [\mathbf{s}_t, \hat{\mathbf{u}}_t]$ is introduced for notational simplicity, which called *system inputs*. It is the collection of system states $\mathbf{s}_t$ and duplicated control actions $\mathbf{u}_t$ and $-\mathbf{u}_t$, therefore ensuring the mapping from $\mathbf{u}_t$ to any future states and outputs remains convex. $J(\hat{\mathbf{x}}_\tau, y_\tau)$ is the control system cost incurs at time $\tau$, that is a function of both the system inputs $\hat{\mathbf{x}}_\tau$ and output $y_\tau$. The functions $f(\cdot)$ and $g(\cdot)$ in Eq. (5b)-(5c) are parameterized as ICRNNs, which represent the system dynamics from sequence of inputs $(\hat{\mathbf{x}}_{\tau-n_w}, \hat{\mathbf{x}}_{\tau-n_w+1}, \ldots, \hat{\mathbf{x}}_\tau)$ to the system output $y_\tau$, and the dynamics from control actions to system states, respectively. $n_w$ is the memory window length of the recurrent neural network. The equations (5d) and (5e) duplicate the input variables $\mathbf{u}$ and enforce the consistency condition between $\mathbf{u}$ and its negation $\mathbf{v}$. Lastly, (5f) and (5g) are the constraints on feasible system states and control actions respectively. Note that as a general formulation, we do not include the duplication tricks on state variables, so the dynamics fitted by (5b) and (5c) are non-decreasing over state space, which are not equivalent to those dynamics represented by linear systems. However, since we are not restricting the control space, and we have explicitly included multiple previous states in the system transition dynamics, so the non-decreasing constraint over state space should not restrict the representation capacity by much. In Section.3 we theoretically prove the representability of proposed networks.

Optimization problem in (5) is a convex optimization with respect to (w.r.t.) inputs $\mathbf{u} = [\mathbf{u}_t, \ldots, \mathbf{u}_{t+T}]$, provided the cost function $J(\hat{\mathbf{x}}_\tau, y_\tau) = J(\mathbf{s}_\tau, \hat{\mathbf{u}}_\tau, y_\tau)$ is convex w.r.t. $\hat{\mathbf{u}}_\tau$, and convex, nondecreasing

w.r.t. $\mathbf{s}_\tau$ and $y_\tau$. A problem is convex if and only if both the objective function and constraints are convex. In the above problem, $J(\mathbf{s}_\tau, \hat{\mathbf{u}}_\tau, y_\tau)$ is convex and nondecreasing w.r.t. $\mathbf{s}_\tau$ and $y_\tau$; $\mathbf{s}_\tau$ and $y_\tau$ are parameterized as ICRNNs, i.e., (5a) and (5b), such that they are convex w.r.t. $\hat{\mathbf{u}}_\tau$. Therefore following the composition rule of convex functions, the objective function is convex w.r.t. inputs $\mathbf{u} = [\mathbf{u}_t, ..., \mathbf{u}_{t+T}]$. Besides, all the equality constraints (5d) and (5e) are affine. Suppose both the state feasibile set (5f) and action feasibile set (5g) are convex, the overall optimization is convex.

The convexity of the problem in (5) guarantees that it can be solved efficiently and optimally using gradient descend method. Since both the objective function (5a) and the constraints (5b)-(5c) are parameterized as neural networks, and their gradients can be calculated via back-propagation with the modification where cost is propagated to the input rather than the weights of the network. For implementation, the gradients can be convinently calculated via existing modules such as Tensorflow viaback-propagation. Let $\mathbf{u}^* = \{\mathbf{u}_t^*, \mathbf{u}_{t+1}^*, ..., \mathbf{u}_{t+T}^*\}$ be the optimal solution of the optimization problem at time $t$. Then the first element of $\mathbf{u}^*$ is implemented to the real-time system control, that is $\mathbf{u}_t^*$. The optimization problem is repeated at time $t+1$, based on the updated state prediction using $\mathbf{u}_t^*$, yielding a model predictive control strategy.

## 3 EFFICIENCY AND REPRESENTATION POWER OF ICNN

Besides the computational traceability of the input convex networks, as an system identification model, we are also interested its predictive accuracies and capacity. This section provides theoretical analysis on the representation ability and efficiency of input convex neural networks.

### 3.1 REPRESENTATION POWER OF INPUT CONVEX NEURAL NETWORK

**Definition 1.** *Given a function $f : \mathbb{R}^d \to \mathbb{R}$, we say that the function $\hat{f}$ approximate $f$ within $\varepsilon$ if $|f(\mathbf{x}) - \hat{f}(\mathbf{x})| \leq \varepsilon$ for all $\mathbf{x}$ in the domain of $f$.*

**Theorem 1.** *[Representation power of ICNN] For any Lipschitz convex function over a compact domain, there exists a neural network with nonnegative weights and ReLU activation functions that approximates it within $\varepsilon$.*

**Lemma 1.** *Given a continuous Lipschitz convex function $f : \mathbb{R}^d \to \mathbb{R}$ with compact domain and $\varepsilon > 0$, it can be approximated within $\varepsilon$ by maximum of a finite number of affine functions. That is, there exists $\hat{f}(\mathbf{x}) = \max_{i=1,...,N}\{\mu_\mathbf{i}^T \mathbf{x} + b_i\}$ such that $|f(\mathbf{x}) - \hat{f}(\mathbf{x})| \leq \varepsilon$ for all $\mathbf{x} \in dom f$.*

*Sketch of proof for Theorem 1.* Supposing Lemma 1 is true, the proof of Theorem 1 boils down to showing that neural network with nonnegative weights and ReLU activation functions can exactly represent a maximum of affine functions. The proof is constructive. We first construct a neural network with ReLU activation functions and both positive and negative weights, then we show that the weights between different layers of the network can be restricted to be nonnegative by a simple duplication trick. Specifically, since the weights in the input layer and passthrough layers in the ICNN can be negative, we simply add a negation of each input variable (e.g. both $\mathbf{x}$ and $-\mathbf{x}$ are given as inputs) to the network. These variables need satisfy a consistency constraint since one is the negation of the other. Since this constraint is linear, it preserves the convexity of optimization problems. The details of the proofs are given in the Appendix B.

This proof is similar in spirit to theorems in (Hanin, 2017; Arora et al., 2016). The key new result is a simpler construction than the one used in (Hanin, 2017) and the restriction to nonnegative weights between the layers. $\qquad\square$

Similar to Theorem 1, an analogous result about the representation power of ICRNN can be shown for systems with convex dynamics. Given a dynamical system described by rolled out system dynamics $y_t = f(\mathbf{x}_1, \ldots, \mathbf{x}_t)$ is convex, then there exists a recurrent neural network with nonnegative weights and ReLU activation functions that approximates it within $\varepsilon$. A broad range of systems can be captured by this model. For example, the linear quadratic (Gaussian) regulator problem can be described using a ICRNN if we identify $y$ as the cost of the regulator (Skogestad & Postlethwaite, 2007; Boyd et al., 1994).[1] An example of a nonlinear system is the control of electrochemical

---

[1]It's important to note that $y$ is usually used as the system output of a linear system, but in our context, we are using it to refer to the quadratic cost with respect to the system states and the control input.

batteries. It can be shown from first principles that the degradation of these types of batteries is convex in their charge and discharge actions (Shi et al., 2018) and our framework offers a powerful data-driven way to control batteries found in electric vehicles, cell phones, and power systems.

## 3.2 ICNN VS. CONVEX PIECEWISE LINEAR FITTING

In the proof of Theorem 1, we first approximate a convex function by a maximum of affine functions then construct a neural network according to this maximum. Then a natural question is why learn a neural network and not directly the affine functions in the maximum? This approach was taken in (Magnani & Boyd, 2009), where a convex piecewise-linear function (max of affine functions) are directly learned from data through a regression problem.

A key reason that we propose to use ICNN (or ICRNN) to fit a function rather than directly finding a maximum of affine functions is that the former is a much more efficient parameterization than the latter. As stated in Theorem 2, a maximum of $K$ affine functions can be represented by an ICNN with $K$ layers, where each layer only requires a single ReLU activation function. However, given a single layer ICNN with $K$ ReLU activation functions, it may take a maximum of $2^K$ affine functions to represent it exactly. Therefore in practice, it would be much easier to train a good ICNN than finding a good set of affine functions.

**Theorem 2.** *[Efficiency of Representation]*

1. *Let $f_{ICNN} : \mathbb{R}^d \rightarrow \mathbb{R}$ be an input convex neural network with K ReLU activation functions. Then $\Omega(2^K)$ functions are required to represent $f_{ICNN}$ using a max of affine functions.*
2. *Let $f_{CPL} : \mathbb{R}^d \rightarrow \mathbb{R}$ be a max of K affine functions. Then $O(K)$ activation functions are sufficient to represent $f_{CPL}$ exactly with an ICNN.*

The proof of this theorem is given in Appendix C.

## 4 EXPERIMENTS

In this section, we verify the effectiveness of ICNN and ICRNN by presenting experimental results on two decision-making problems: continuous control benchmarks on MuJoco locomotion tasks (Todorov et al., 2012) and energy management of reference large-scale commercial building (Crawley et al., 2001), respectively. The proposed method can be used as a flexible building block in decision making problems, where we use ICNN to represent system dynamics for MuJoco simulators, and we use ICRNN in an end-to-end fashion to find the optimal control inputs. Both examples demonstrate that proposed method: 1) discovers the connection between controllable variables and the system dynamics or cost objectives; 2) is lightweight and sample-efficient; 3) achieves generalizable and more stable control performances compared with previous model-based reinforcement learning and simplified linear control approaches.

## 4.1 MUJOCO LOCOMOTION TASKS

**Experimental Setup** We consider four simulated robotic locomotion tasks: `swimmer`, `half-cheetah`, `hopper`, `ant` implemented in MuJoCo under the OpenAI rllab framework (Duan et al., 2016). We train and represent the locomotion state transition dynamics $\mathbf{s}_{t+1} = g(\mathbf{s}_t, \mathbf{u}_t)^2$ using a 2-layer ICNN with ReLU activations, which could be integrated into the following finite-horizon control problem to find the optimal action sequence $\mathbf{u}_t, ..., \mathbf{u}_{t+T}$ for fixed looking ahead horizon $T$:

$$\underset{\mathbf{u}_t,...,\mathbf{u}_{t+T}}{\text{minimize}} \quad -\sum_{\tau=t}^{t+T} r(\mathbf{s}_\tau, \mathbf{u}_\tau) \tag{6a}$$

$$\text{subject to} \quad \mathbf{s}_{\tau+1} = g(\mathbf{s}_\tau, \mathbf{u}_\tau), \forall \tau \in [t, t+T] \tag{6b}$$

$$\mathbf{u}_\tau \in \mathscr{U}_{feasible}, \forall \tau \in [t, t+T] \tag{6c}$$

---

[2]Note that for notation convenience, in this example and the following building example, we use $\mathbf{u}$ to represent the expanded control vector including its negation. For system state $\mathbf{s} \in R^d$, if $d > 1$, convexity means that each dimension of $\mathbf{s}$ is convex w.r.t. the function inputs.

where the objective (6a) is convex because $r(\mathbf{s}_\tau, \mathbf{u}_\tau)$ is a concave reward function related to system states such as velocity and control actions (the detailed forms of $r(\mathbf{s}_\tau, \mathbf{u}_\tau)$ for different locomotion tasks are listed in Appendix D). To achieve better model generalization on locomotion dynamics, we also followed (Nagabandi et al., 2018), and applied DAGGER (Ross et al., 2011) to iteratively collect labeled robotic rollouts and train the supervised dyamics model (6b) using on-policy locomotion samples. See Appendix D for furthur simulation hyperparameters and experimental details. For each aggregated iterations of collecting rollouts data and training ICNN model, we validate the controller performance on standalone validation rollouts by optimally solving (6).

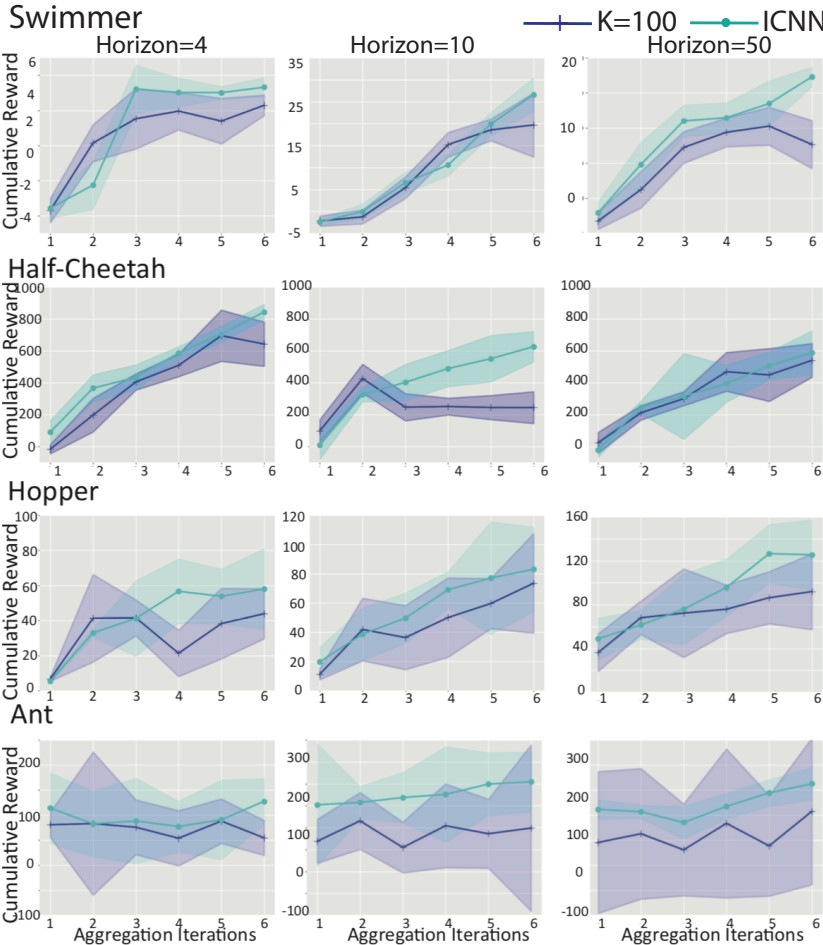

Figure 3: Average rollout reward for random-shooting method *vs* ICNN on four MuJoCo tasks. The horizontal axis indicates the aggregated iteration, and vertical axis indicates average reward. Plotted curves are averaged over 3 random seeds, and the shaded region shows the standard deviation.

**Baselines** We compare our system modeling and continuous control method with state-of-the-art model-based RL algorithm (Nagabandi et al., 2018), where the authors used a normal multi-layer perceptrons (MLP) model to parameterize the system dynamics (6b). We refer to their method as random-shooting algorithm, since they can not solve (6) to optimality, and they used pre-defined number of random-shooting control sequences (denoted as $K$) to query the trained MLP and find a best sequence as the rollout policy. Such a method is able to find good control policies in the degree of $10^4$ timesteps, which are much more sample-efficient than model-free RL methods (Duan et al., 2016; Mnih et al., 2015). To make fair comparisons with baseline method, we keep the same setup on the rollouts number and initial random action training. Our framework makes the neural networks convex w.r.t input by adding passthrough links to the 2-layer model and keeping all the layer weights nonnegative. We evaluate the performance of both algorithms on three randomly selected fixed random seeds for four tasks. Similar to the fine tuning steps in (Nagabandi et al., 2018), control

policies found by ICNN can also be plugged in as initialized policies for subsequent model-free reinforcement learning algorithms.

**Continuous Control Performance**   During training, we found both ICNN and MLP are able to predict robotic states quite accurately based on (6b). This provides a good system dynamics model which is beneficial to solve control policies. The control performances are shown in Fig. 3, where we compare the average reward of proposed method and random-shooting method with $K = 100$ over 10 validation rollouts during each aggregated iteration (see Fig. 8 in Appendix D.4 for random shooting performance with varying $K$). The policy found by ICNN outperforms the random-shooting method in all settings with varying horizon $T$ for all of the four locomotion tasks.

Intuitively, ICNN should perform better when the action space is larger, since random-shooting method can not search through the action space efficiently with a fixed $K$. This is illustrated in the example of `ant`, where with more training samples aggregated and MLP model representing more accurate dynamics, random-shooting gets stuck to find better control policies and there is little improvement reflected in the control performance. Moreover, since we are skipping the expensive process on calculating rewards of each random shooting trajectory and finding the best one, our method only implements ICNN inference step based on (6) and is much faster than random shooting methods in most settings, especially when $K$ is large (see Table. 2 for wall-clock time in Appendix D.3). For instance, in the case of `Swimmer`, our proposed method only uses $\frac{1}{5}$ of time compared to (Nagabandi et al., 2018). This also indicates that our method is even much more sample-efficient than off-the-shelf model-free RL methods, where we use two orders of magnitude less training data to reach similar validation rewards (Duan et al., 2016; Mnih et al., 2015) (see Fig. 9 in Appendix D.4).

## 4.2   BUILDING ENERGY MANAGEMENT

**Experimental Setup**   We now move on to optimally control a dynamical system with significant inertia. We consider the real-time control problem of building's HVAC (heating, ventilation, and air conditioning) system to reduce its energy consumption. Building energy management remains to be a hard problem in control area. The exact system dynamics are unknown and hard to model due to the complex heating transfer dynamics, time-varying environments and the scale of the system in terms of states and actions (Kouro et al., 2009). At time $t$, we assume the building's running profile $\mathbf{x}_t := [\mathbf{s}_t, \mathbf{u}_t]$ is available, where $\mathbf{s}_t$ denotes building system states, including outside temperature, room temperature measurements, zone occupancies and etc. $\mathbf{u}_t$ denotes a collection of control actions such as room temperature set points and appliance schedule. Output is the electricity consumption $P_t$.

This is a model predictive control problem in the sense that we want to find the best control inputs that minimize the overall energy consumption of building by looking ahead several time steps. To achieve this goal, we firstly learn an ICRNN model $f(\cdot)$ of the building dynamics, which is trained to minimize the error between $P_t$ and $f(\mathbf{x}_{t-n_w}, ..., \mathbf{x}_t)$, while $n_w$ denotes the memory window of recurrent neural networks. Then we solve:

$$\underset{\mathbf{u}_t,...,\mathbf{u}_{t+T}}{\text{minimize}} \quad \sum_{\tau=t}^{t+T} f(\mathbf{x}_{\tau-n_w}, ..., \mathbf{x}_\tau) \tag{7a}$$

$$\text{subject to} \quad \mathbf{s}_\tau = g(\mathbf{x}_{\tau-n_w}, ..., \mathbf{x}_{\tau-1}, \mathbf{u}_\tau), \forall \tau \in [t, t+T] \tag{7b}$$

$$\underline{\mathbf{u}}_\tau \leq \mathbf{u}_\tau \leq \overline{\mathbf{u}}_\tau, \forall \tau \in [t, t+T] \tag{7c}$$

$$\underline{\mathbf{s}}_\tau \leq \mathbf{s}_\tau \leq \overline{\mathbf{s}}_\tau, \forall \tau \in [t, t+T] \tag{7d}$$

where the objective (7a) is minimizing the total energy consumption in future $T$ steps ($T$ is the model predictive control horizon), and (7b) is used for modeling building states, in which $g(\cdot)$ are parameterized as ICRNNs. Note that the formulation (7) is also flexible with different loss functions. For instance, in practice, we could reuse trained dynamics model (7b), and integrate electricity prices into the overall objective so that we could directly learn real-time actions to minimize electricity bills (please refer to Appendix E for more results). The constraints on control actions $\mathbf{u}_t$ and system states $\mathbf{s}_t$ are given in (7c) and (7d). For instance, the temperature set points as well as real measurements should not exceed user-defined comfort regions.

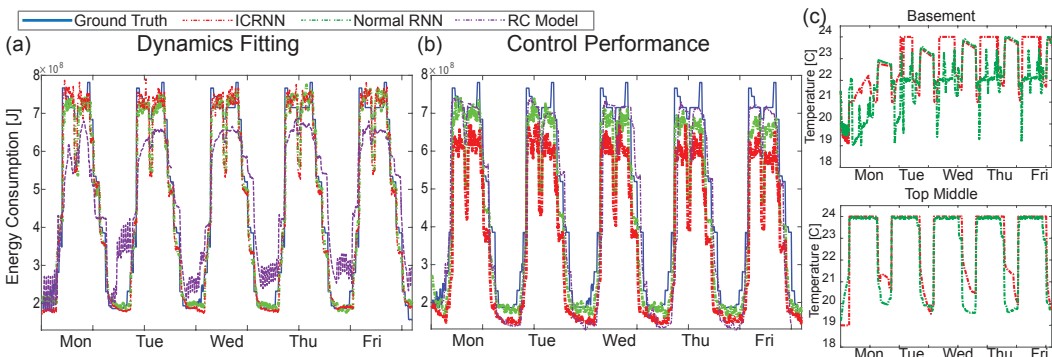

Figure 4: Results for constrained optimization of building energy management. (a) ICRNN is able to model the building dynamics as accurately as conventional RNN; (b) Compared to conventional RNN model, ICRNN finds control actions which lead to 11.52% more of energy savings, and (c) ICRNN provides stable control actions while decisions generated by conventional RNN vary dramatically.

To test the performance of the proposed method, we set up a 12-story large office building, which is a reference EnergyPlus commercial building model from US Department of Energy (DoE) [3], with a total floor area of 498, 584 square feet which is divided into 16 separate zones. By using the whole year's weather profile, we simulate the building running through the year and record $(\mathbf{x}_t, P_t)$ with a resolution of 10 minutes. We use 10 months' data to train the ICRNN and subsequent 2 months' data for testing. We use 39 building system state variables $\mathbf{s}_t$ (uncontrollable), along with 16 control variables $\mathbf{u}_t$. Output is a single value of building energy consumption at each time step. We set the model predictive control horizon $T = 36$ (six hours). We employ an ICRNN with recurrent layer of dimension 200 to fit the building input-output dynamics $f(\cdot)$. The model is trained to minimize the MSE between its predictions and the actual building energy consumption using stochastic gradient descent. We use the same network structure and training scheme to fit state transition dynamics $g(\cdot)$.

**Baseline** We set the model-based forecasting and optimization benchmark using an linear resistor-circuit (RC) circuit model to represent the heat transfer in building systems, and solve for the optimal control actions via MPC (Ma et al., 2012). At each step, MPC algorithm takes into account the forecasted states of the building based on the fitted RC model and implements the current step control actions. We also compare the performance of ICRNN against the conventionally trained RNN in terms of building dynamics fitting performance and control performance. To solve the MPC problem with conventional RNN models, we also use gradient-based method with respect to controls. However, since conventional RNN models are generally not convex from input to output, there is no guarantee to reach a global optimum (or even a local one).

**Results** In terms of the fitting performance, ICRNN provides a competitive result compared to conventional RNN model. The overall test root mean square error (RMSE) is 0.054 for ICRNN and 0.051 for conventional RNN, both of which are much smaller than the error made by RC model (0.240). Fig. 4(a) shows the fitting performance on 5 working days in test data. This illustrates the good performance of ICRNN in modeling building HVAC system dynamics. Then by using the learned ICRNN model of building dynamics, we obtain the suggested room control actions $u_t^*$ by solving the optimal building control problem (7). As shown in Fig. 4(b), with the same constraints on building temperature interval of $[19°C, 24°C]$, the building energy consumption is reduced by 23.25% after implementing the new temperature set points calculated by ICRNN. On the contrary, since there is no guarantee for finding optimal control actions by optimizing over conventional RNN's input, the control solutions given by conventional RNN could only reduce 11.73% of electricity. Solutions given by RC model only saves 4.07% of electricity. More importantly, in Fig. 4(c) we demonstrate the control actions outputted by our method against MPC with conventional RNN in two randomly selected building zones, the building basement and top floor central area. It shows that our proposed

---

[3]Energyplus is an open-source whole-building energy modeling software, which is developed by US DoE for standard building energy simulation

approach is able to find a group of stable control actions for the building system control. While in the conventional RNN case, it generates control set points which have undesirable, drastic variations.

## 5 SUMMARY AND DISCUSSION

In this work we proposed a novel optimal control framework that uses deep neural networks engineered to be convex from the input to the output. This framework bridges machine learning and control by representing system dynamics using input convex (recurrent) neural networks. We show that many interesting data-driven control problems can be cast as convex optimization problems using the proposed network architecture. Experiments on both benchmark MuJoCo locomotion tasks and building energy management demonstrate our methodology's potential in a variety of control and optimization problems.

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

APPENDIX

A. TOY EXAMPLE

Consider a synthetic example which contains two circles of noisy input data $\mathbf{u} \in \mathbb{R}^2$, along with discrete data label $y \in \{0,1\}$ which is based on input coming from inner loop ($y = 0$) or outer loop ($y = 1$). Suppose a decision maker is interested in finding the $\mathbf{u}$ that maximizes the probability of $y$ being 0. This optimization problem can be solved by firstly learning a neural network classifier from $\mathbf{u}$ to $y$, and then to find the $\mathbf{u}$ point which minimizes the output of the neural network. More specifically, let $f_{NN}$ be a conventional neural network and $f_{ICNN}$ be an ICNN. Then the objective becomes minimizing $f_{NN}(\mathbf{u})$ or $f_{ICNN}(\mathbf{u})$.

Figure 5 shows the decision boundaries for $f_{NN}$ and $f_{ICNN}$, respectively. These networks are composed of 2 hidden layers, with 200 neurons in each layer, and are trained using the same random seed, same number of samples (100) until loss convergence. The decision boundaries of a conventional network have many "zigzags", which makes solving (1) challenging, especially if $\mathbf{u}$ is constrained. In contrast, the ICNN has convex level sets (by construction) as decision boundaries, which leads to a convex optimization problem.

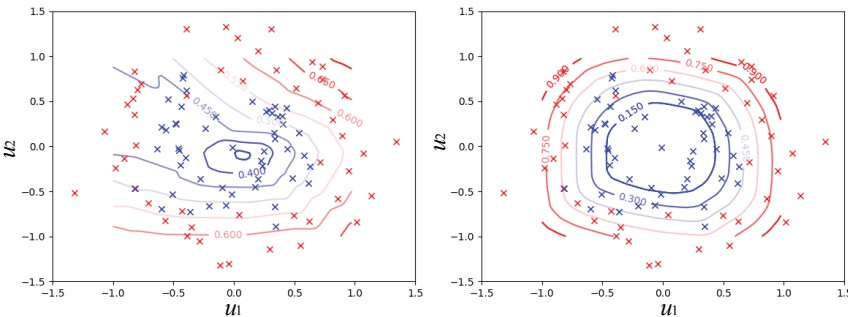

Figure 5: Toy example on classifying circle data with label 0 (blue cross) and label 1 (red cross) along with conventional neural networks (left) and ICNN (right) decision contour lines. A decision maker is interested in finding a $\mathbf{u}$ that has the highest probability of being labeled 0.

APPENDIX B. PROOF OF THEOREM 1

*Proof.* Lemma 1 follows from well established facts in function analysis stating that piecewise linear functions are dense in the space of all continuous functions over compact sets (Royden & Fitzpatrick, 2010) and convex piecewise linear functions are dense in the space of all convex continuous functions (Cox, 1971; Gavrilović, 1975). Using the fact that convex piecewise linear functions can be represented as a maximum of affine functions (Magnani & Boyd, 2009; Wang, 2004) gives the desired result in the lemma.

Lemma 1 shows that all continuous Lipschitz convex functions $f(\mathbf{x}) : \mathbb{R}^d \to \mathbb{R}$ over convex compact sets can be approximated using maximum of affine functions. Then it suffices to show that an ICNN can exactly represent a maximum of affine functions. To do this, we first construct a neural network with ReLU activation function with both positive and negative weights that can represent a maximum of affine functions. Then we show how to restrict all weights to be nonnegative.

As a starting example, consider a maximum of two affine functions

$$f_{CPL}(\mathbf{x}) = \max\{\mathbf{a}_1^T \mathbf{x} + b_1, \mathbf{a}_2^T \mathbf{x} + b_2\}. \tag{8}$$

To obtain the exact same function using a neural network, we first rewrite it as

$$f_{CPL}(x) = (\mathbf{a}_2^T \mathbf{x} + b_2) + \max\left((\mathbf{a}_1 - \mathbf{a}_2)^T \mathbf{x} + (b_1 - b_2), 0\right). \tag{9}$$

Now define a two-layer neural network with layers $\mathbf{z}_1$ and $\mathbf{z}_2$ as shown in Fig. 6:

$$z_1 = \sigma\left((\mathbf{a}_1 - \mathbf{a}_2)^T \mathbf{x} + (b_1 - b_2)\right), \tag{10a}$$

$$z_2 = z_1 + \mathbf{a}_2^T \mathbf{x} + b_2 \tag{10b}$$

where $\sigma$ is the ReLU activation function and the second layer is linear. By construction, this neural network is the same function as $f_{CPL}$ given in (8).

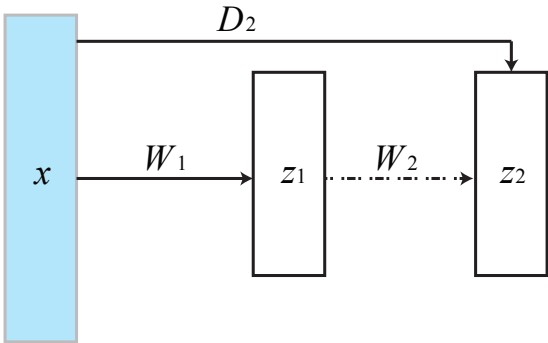

Figure 6: A simple two-layer neural networks. In alignment with (10), $W_1$ denotes the first-layer weights $\mathbf{a}_1 - \mathbf{a}_2$ and bias $b_1 - b_2$, and $W_2$ denotes the linear second layer. Direct layer is denoted as $D_2$ for weights $\mathbf{a}_2$ and bias $b_2$.

The above argument extends directly to a maximum of $K$ linear functions. Suppose

$$f_{CPL}(\mathbf{x}) = \max\{\mathbf{a}_1^T\mathbf{x} + b_1, ..., \mathbf{a}_K^T\mathbf{x} + b_K\} \tag{11}$$

Again the trick is to rewrite $f_{CPL}(\mathbf{x})$ as a nested maximum of affine functions. For notational convenience, let $L_i = \mathbf{a}_i^T\mathbf{x} + b_i$, $L_i' = L_i - L_{i+1}$. Then

$$
\begin{aligned}
f_{CPL} &= \max\{L_1, L_2, ..., L_K\} \\
&= \max\{\max\{L_1, L_2, ..., L_{K-1}\}, L_K\} \\
&= L_K + \sigma\left(\max\{L_1, L_2, ..., L_{K-1}\} - L_K\right) \\
&= L_K + \sigma\left(\max\{\max\{L_1, L_2, ..., L_{K-2}\}, L_{K-1}\} - L_K, 0\right) \\
&= L_K + \sigma\left(L_{K-1} - L_K + \sigma\left(\max\{L_1, L_2, ..., L_{K-2}\} - L_{K-1}, 0\right), 0\right) \\
&= ... \\
&= L_K + \sigma\left(L_{K-1}' + \sigma\left(L_{K-2}' + \sigma\left(...\sigma\left(L_2' + \sigma\left(L_1 - L_2, 0\right), 0\right), ..., 0\right), 0\right), 0\right).
\end{aligned}
$$

The last equation describes a $K$ layer neural network, where the layers are:

$$
\begin{aligned}
z_1 &= \sigma\left(L_1 - L_2, 0\right) = \sigma\left((\mathbf{a}_1 - \mathbf{a}_2)^T\mathbf{x} + (b_1 - b_2)\right), \\
z_2 &= \sigma\left(L_2' + z_1, 0\right) = \sigma\left(z_1 + (\mathbf{a}_2 - \mathbf{a}_3)^T\mathbf{x} + (b_2 - b_3)\right), \\
&\quad ...... \\
z_i &= \sigma\left(L_i' + z_{i-1}, 0\right) = \sigma\left(z_{i-1} + (\mathbf{a}_i - \mathbf{a}_{i+1})^T\mathbf{x} + (b_i - b_{i+1})\right), \\
&\quad ...... \\
z_K &= z_{K-1} + L_K = h_K\left(z_{K-1} + L_K\right) = \left(z_{K-1} + \mathbf{a}_K^T\mathbf{x} + b_K\right).
\end{aligned}
$$

Each layer of of this neural network uses only a single activation function.

Although the above neural network exactly represent a maximum of linear functions, it is not convex since the coefficients between layers could be negative. In particular, each layer involves an inner product of the form $(\mathbf{a}_i - \mathbf{a}_{i+1})^T\mathbf{x}$ and the coefficients are not necessarily nonnegative. To overcome this, we simply expand the input to include $\mathbf{x}$ and $-\mathbf{x}$. Namely, define a new input $\hat{\mathbf{x}} \in \mathbb{R}^{2d}$ as

$$\hat{\mathbf{x}} = \begin{bmatrix} \mathbf{x} \\ -\mathbf{x} \end{bmatrix}. \tag{12}$$

Then any inner product of the form $\mathbf{h}^T \mathbf{x}$ can be written as

$$
\begin{aligned}
\mathbf{h}^T \mathbf{x} &= \sum_{j=1}^{d} h_i x_i \\
&= \sum_{i:h_i \geq 0} h_i x_i + \sum_{i:h_i < 0} h_i x_i \\
&= \sum_{i:h_i \geq 0} h_i x_i + \sum_{i:h_i < 0} (-h_i)(-x_i) \\
&= \sum_{i:h_i \geq 0} h_i \hat{x}_i + \sum_{i:h_i < 0} (-h_i)(\hat{x}_{i+d}),
\end{aligned}
$$

where all coefficients are nonnegative in the above sum.

Therefore any inner product between a coefficient vector and the input $\mathbf{x}$ can be written as an inner product between a nonnegative coefficient vector and the expanded input $\hat{\mathbf{x}}$. Therefore, without loss of generality, we can limit all of the weights between layers to be nonnegative, and thus the neural network to be input convex. Note that in optimization problems, we need to enforce consistency in $\hat{\mathbf{x}}$ be including (12) as a constraint. However, this is a linear equality constraint, which maintains the convexity of the optimization problem.

$\square$

APPENDIX C. PROOF OF THEOREM 2

*Proof.* The second statement of Theorem 2 directly follows the construction in the proof of Theorem 1, which shows that a maximum of $K$ affine functions can be represent by a $K$-layer ICNN (with a single ReLU function in each layer). So it remains to show the first statement of Theorem 2.

To show that a maximum of affine functions can require exponential number of pieces to approximate a function specified by an ICNN with $K$ activation functions, consider a network with 1 hidden layer of K nodes and the weights of direct "passthrough" layers are set to 0:

$$
f_{ICNN}(\mathbf{x}) = \sum_{i=1}^{K} w_{1i} \sigma(\mathbf{w}_{0i}^T \mathbf{x} + b_i), \tag{13}
$$

It contains $3K$ parameters: $\mathbf{w}_{0i}$, $w_{1i}$ and $b_i$, where $\mathbf{w}_{0i} \in \mathbb{R}^d$ and $w_{1i}, b_i \in \mathbb{R}$.

In order to represent the same function by a maximum of affine functions, we need to assess the value of every activation unit $\sigma(\mathbf{w}_{0i}^T \mathbf{x} + b_i)$. If $\mathbf{w}_{0i}^T \mathbf{x} + b_i \geq 0$, $\sigma(\mathbf{w}_{0i}^T \mathbf{x} + b_i) = \mathbf{w}_{0i}^T \mathbf{x} + b_i$; otherwise, $\sigma(\mathbf{w}_{0i}^T \mathbf{x} + b_i) = 0$. In total, we have $2^K$ potential combinations of piecewise-linear function, including

$$
\begin{aligned}
L_1 &= \left( \sum_{i=1}^{K} w_{1i} \mathbf{w}_{0i} \right)^T \mathbf{x} + \sum_{i=1}^{K} w_{1i} b_i, \text{if all } \mathbf{w}_{0i}^T \mathbf{x} + b_i \geq 0 \\
L_2 &= \left( \sum_{i=2}^{K} w_{1i} \mathbf{w}_{0i} \right)^T \mathbf{x} + \sum_{i=2}^{K} w_{1i} b_i, \text{if } \mathbf{w}_{01}^T \mathbf{x} + b_1 < 0 \text{ and all other } \mathbf{w}_{0i}^T \mathbf{x} + b_i \geq 0 \\
L_3 &= \left( w_{11} \mathbf{w}_{01} + \sum_{i=3}^{K} w_{1i} \mathbf{w}_{0i} \right)^T \mathbf{x} + w_{1i} b_i + \sum_{i=3}^{K} w_{1i} b_i, \text{if } \mathbf{w}_{02}^T \mathbf{x} + b_2 < 0 \text{ and other } \mathbf{w}_{0i}^T \mathbf{x} + b_i \geq 0 \\
& \quad \ldots \ldots, \\
L_{2^K} &= 0, \text{ if all } \mathbf{w}_{0i}^T \mathbf{x} + b_i < 0.
\end{aligned}
$$

So the following maximum over $2^K$ pieces is required to represent the single linear ICNN:

$$
\max\{L_1, L_2, ..., L_{2^K}\}.
$$

$\square$

| Environment | Swimmer | Half-Cheetah | Hopper | Ant |
|---|---|---|---|---|
| Reward Function | $s_{t+1}^{vel} - 0.5\|\|\frac{\mathbf{u}}{50}\|\|_2^2$ | $s_{t+1}^{vel} - 0.05\|\|\frac{\mathbf{u}}{1}\|\|_2^2$ | $s_{t+1}^{vel} + 1 - 0.005\|\|\frac{\mathbf{u}}{200}\|\|_2^2$ | $s_{t+1}^{vel} + 0.5 - 0.005\|\|\frac{\mathbf{u}}{150}\|\|_2^2$ |
| Rollout Horizon | 333 | 1000 | 200 | 1000 |
| Rollout Numbers | 25 | 10 | 30 | 400 |
| Training Epochs | 60 | 60 | 40 | 60 |

Table 1: Environment and training details for four MuJoCo locomotion tasks.

## APPENDIX D. EXPERIMENTAL DETAILS ON MUJOCO TASKS

### D.1 DATA COLLECTION

**Rollout Samples**    To train the neural network dynamics model (both ICNN and MLP), we first collect initial rollout data using fully random action sequences $\mathbf{u}_t \sim$ Uniform$[\mathbf{-1}, \mathbf{1}]$ with a random chosen initial state. During the data collection process in aggregated iterations, to improve model generalization and explore larger state spaces, we add Gaussian noise to the optimal control policies $\mathbf{u}_t = \mathbf{u}_t + \mathcal{N}(0, 0.001)$.

**Neural Networks Training** We represent the MuJoCo dynamics with a 2-hidden-layer neural networks with hidden sizes $512 - 512$. The passthrough links of ICNN are of same size of corresponding added layers. We train both models using Adam optimizer with a learning rate 0.001 and a mini-batch size of 512. Due to the different complexity of MuJoCo tasks, we vary training epochs and summarize the training details in Table. 1.

### D.2 ENVIRONMENT DETAILS

In all of the MuJoCo locomotion tasks, $\mathbf{s}$ includes state variables such as robot positions, velocity along each axis; $\mathbf{u}$ includes action efforts for the agent. We use standard reward functions $r(\mathbf{s}_t, \mathbf{u}_t)$ for moving tasks, which could be also promptly calculated in (6a) as the control objective. For the ease of neural network training and action sampling, we normalize all the action and states in the range of $[-\mathbf{1}, \mathbf{1}]$. We use DAGGER (Ross et al., 2011) for 6 aggregated iterations for all cases, and during aggregated iteration, we use a split of 10% random rollouts collected as described in 5, and other 90% coming from past iterations' control policies (on-policy rollouts). Note that we use 10 random control sequences in our method to initialize the policy finding approach and avoid the long computation time for taking gradients on finding optimal $\mathbf{u}_t$. Other environment parameters are described in Table. 1.

### D.3 WALL-CLOCK TIME

In Table.2, we show the average run time for the total of 6 aggregation iterations over 3 runs. Finding control policies via ICNN is using less or equal training time compared to random-shooting method with $K = 100$, while achieving better task rewards than $K = 1000$ for different control horizons. All the experiments are running on a computer with 8 cores Intel I7 6700 CPU. Note that we do not use GPU for accelerating ICNN optimization step (6), which could furthur improve our method's efficiency.

### D.4 DETAILS OF SIMULATION RESULTS

**MuJoCo Dynamics Modeling**    In Fig. 7, we compare the ICNN and normal MLP fitting performance of the MuJoCo dynamics modeling (6b), which illustrates that both MLP and ICNN are able to find a data-driven dynamics model for `ant` MuJoCo agent, which is of the most complex dynamics we considered for locomotion tasks. The multi-step prediction errors of ICNN is comparable to normal MLP used in (Nagabandi et al., 2018) for different length of rollout steps.

|  | Swimmer | | | |
|  | $K = 100$ | $K = 300$ | $K = 1000$ | ICNN |
|---|---|---|---|---|
| $H = 4$ | 18.36 | 18.48 | 40.20 | **16.41** |
| $H = 10$ | 21.74 | 25.41 | 71.49 | **18.71** |
| $H = 50$ | 40.01 | 70.31 | 169.49 | **36.24** |
|  | Half-Cheetah | | | |
|  | $K = 100$ | $K = 300$ | $K = 1000$ | ICNN |
| $H = 4$ | **34.40** | 47.72 | 88.49 | 34.93 |
| $H = 10$ | 48.86 | 74.60 | 181.34 | **36.39** |
| $H = 50$ | 113.58 | 275.61 | 816.32 | **83.66** |
|  | Hopper | | | |
|  | $K = 100$ | $K = 300$ | $K = 1000$ | ICNN |
| $H = 4$ | **5.48** | 6.30 | 7.76 | 5.61 |
| $H = 10$ | 5.97 | 7.89 | 9.34 | **5.14** |
| $H = 50$ | 10.89 | 14.77 | 38.02 | **9.16** |
|  | Ant | | | |
|  | $K = 100$ | $K = 300$ | $K = 1000$ | ICNN |
| $H = 4$ | 399.39 | 415.51 | 433.35 | **349.13** |
| $H = 10$ | 480.60 | 481.34 | 511.93 | **459.63** |
| $H = 50$ | 979.73 | 1024.5 | 1075.52 | **929.5** |

Table 2: Average wall clock time (in minutes) for random-shooting model-based reinforcement learning method and ICNN.

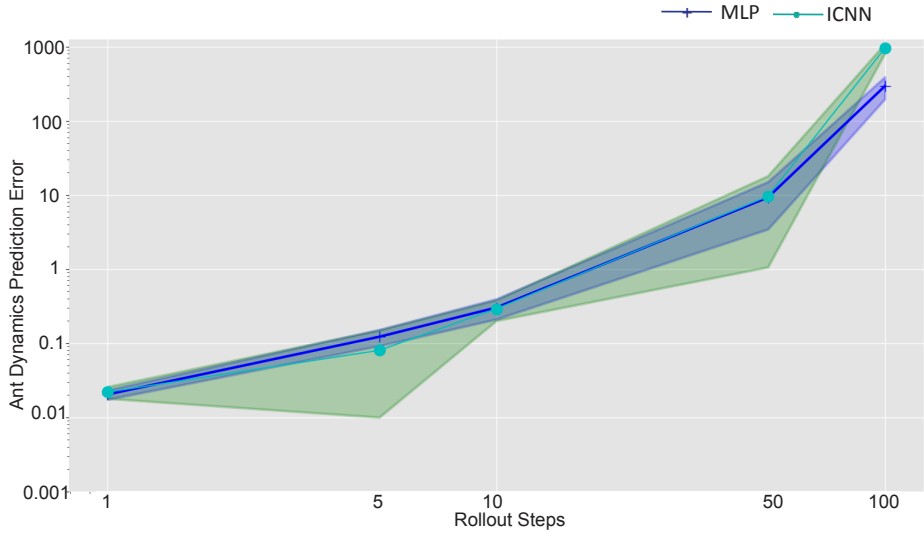

Figure 7: Multistep prediction errors by ICNN and MLP. X-Axis and Y-Axis are of log scale.

**More Simulation Results** In Fig. 8, we compare our control method with random-shooting approach with varying settings on shooting number $K$, which shows that our approach is more efficient in finding control policies.

In Fig. 9, we compare our control method with the rllab implementation of trust region policy optimization (TRPO) (Schulman et al., 2015), an end-to-end deep reinforcement learning approach for mujoco locomotion tasks. More specifically, we compare the algorithms' performances with relatively few available rollout samples. While our approach quickly learns the dynamics and then find control actions via optimization steps, TRPO is hard to learn the actions directly with few provided rollouts. Similarly to the model-based and model-free (Mb-Mf) approach described

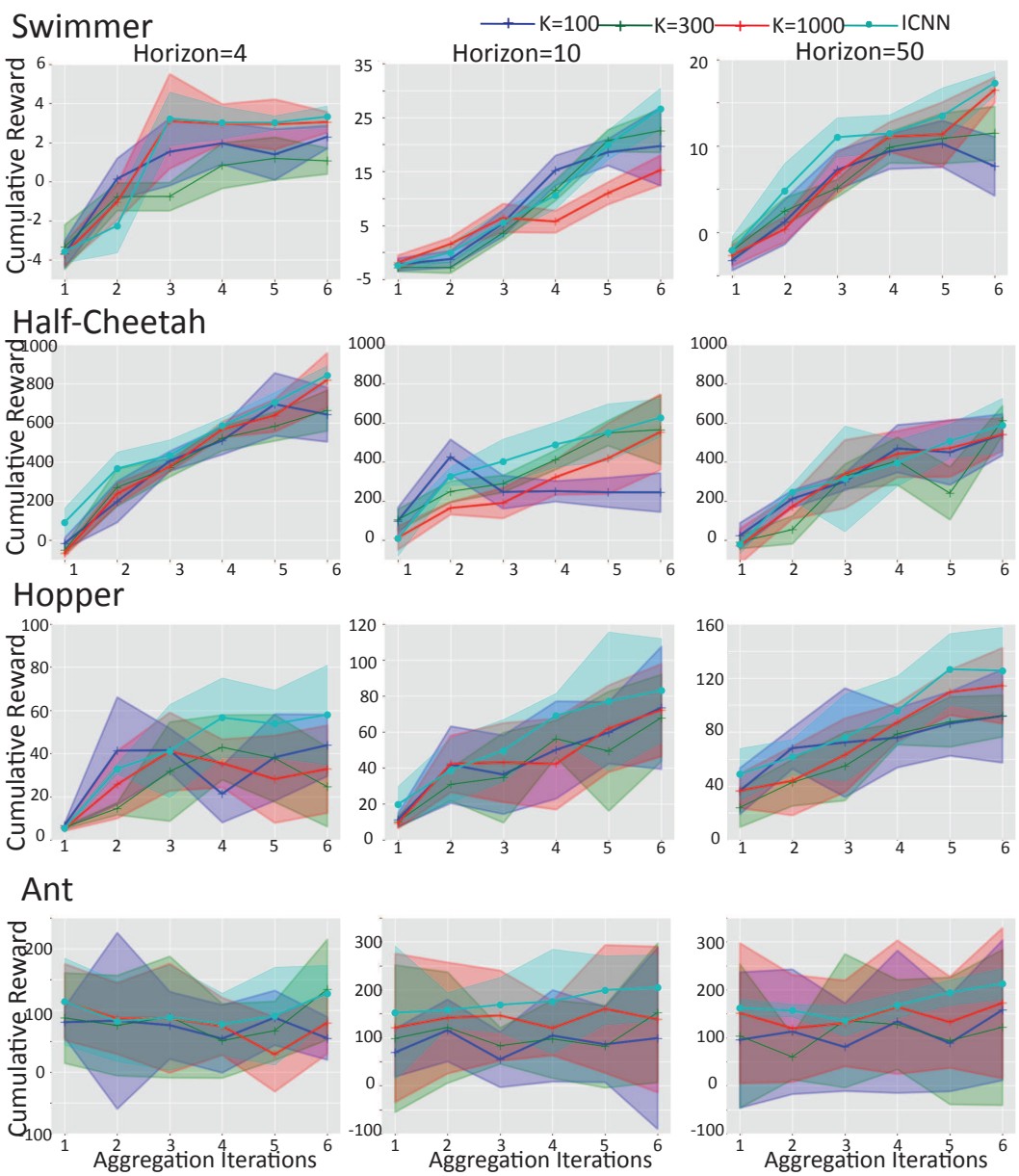

Figure 8: Cumulative reward for one validation rollout of random shooting method *vs* ICNN

in (Nagabandi et al., 2018), our control method could provide good initialization samples for the model-free algorithms, which could greatly accelerate the training process of model-free algorithms.

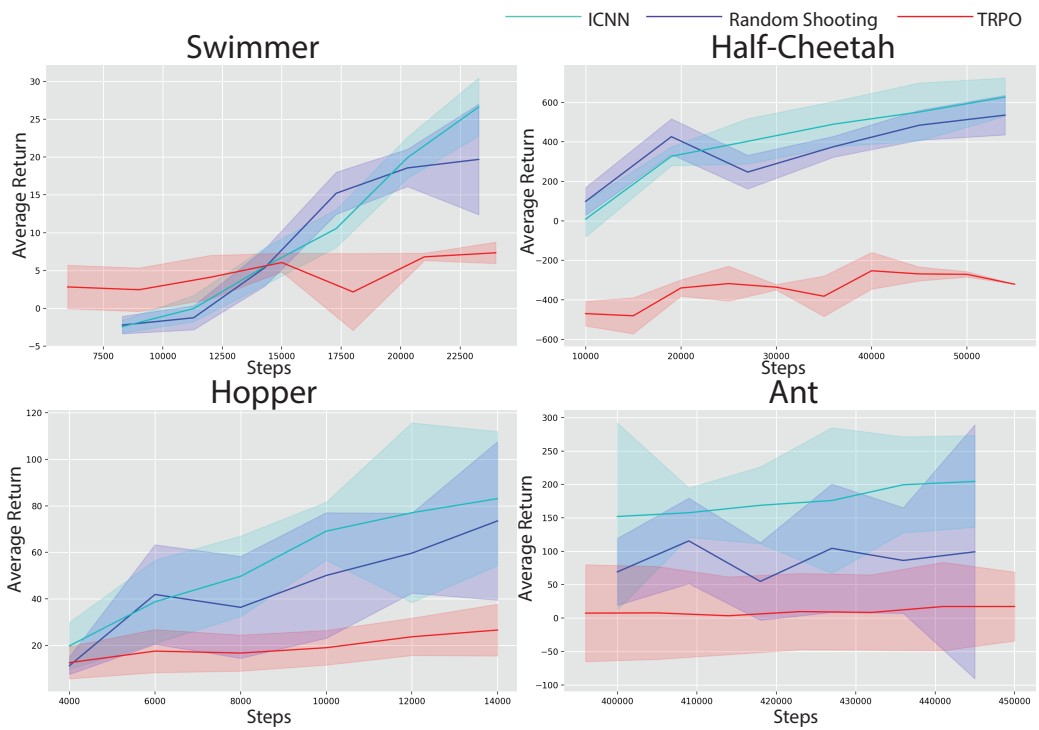

Figure 9: Average return for control of Mujoco tasks by ICNN, random-shooting method (Nagabandi et al., 2018) and TRPO (Schulman et al., 2015).

APPENDIX E. DETAILS ON BUILDING ENERGY MANAGEMENT

E.1 MINIMIZING ELECTRICITY COSTS

To further demonstrate the potential of our proposed control framework in dealing with different real world tasks, we modify the setting of the building control example in Section 4.2 to a more complicated case. Instead of directly minimize the total energy consumption of building, we aim to minimize the total energy cost of building which subject to a varying time-of-use electrical price $\lambda$. The optimization problem in (7) should be re-written as,

$$\underset{\mathbf{u}_t,...,\mathbf{u}_{t+T}}{\text{minimize}} \quad \sum_{\tau=0}^{T} \lambda_\tau \cdot f(\mathbf{x}_{t+\tau-n_w},...,\mathbf{x}_{t+\tau}) \tag{14a}$$

$$\text{subject to} \quad \mathbf{s}_{t+\tau} = g(\mathbf{x}_{t+\tau-n_w},...,\mathbf{x}_{t+\tau-1},\mathbf{u}_{t+\tau}),\forall \tau \tag{14b}$$

$$\underline{\mathbf{u}}_{t+\tau} \leq \mathbf{u}_{t+\tau} \leq \overline{\mathbf{u}}_{t+\tau},\forall \tau \tag{14c}$$

$$\underline{\mathbf{s}}_{t+\tau} \leq \mathbf{s}_{t+\tau} \leq \overline{\mathbf{s}}_{t+\tau},\forall \tau \tag{14d}$$

where the objective (14a) is minimizing the total energy cost of building in future $T$ steps ($T$ is the model predictive control horizon) subject to time-of-use electricity price $\lambda_\tau$, and (14b) is used for modeling building states, in which $g(\cdot)$ are parameterized as ICRNNs. Same as the previous building control case, we have constraints on both control actions $\mathbf{u}_t$ and system states $\mathbf{s}_t$ are given in (14c) and (14d). For instance, the temperature set points as well as real measurements should not exceed user-defined comfort regions. In Fig. 10 we visualize our model flexibility by using Seattle's Time-of-Use (TOU) price from Seattle City Light [4], and minimizing one week's electricity bills. We could see ICRNN capture the long term relationships between control variables and final costs, and raise the energy consumption during off-peak price a little, but reduce the energy consumption during peak hours.

---

[4]http://www.seattle.gov/light/

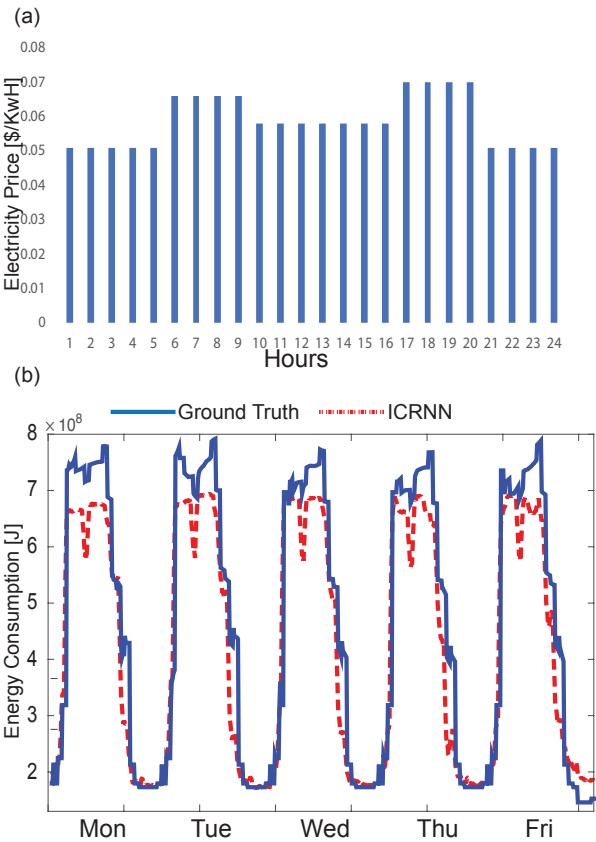

Figure 10: (a) 24 hour price signal along with (b) optimization results on one-week electricity usage of building using ICRNN.

E.2 CONTROL CONSTRAINTS EFFECTS

In Fig. 11 we add one more comparison on the control constraints effects on the final control performance by using ICRNN. Interestingly, with different set point constraints, the ICRNN finds similar solutions for off-peak electricity usage, which may correspond to necessary energy consumptions, such as lightning and ventilation. Moreover, when we set no constraints on the system, it would cut down more than 80% of total energy during peak hours.

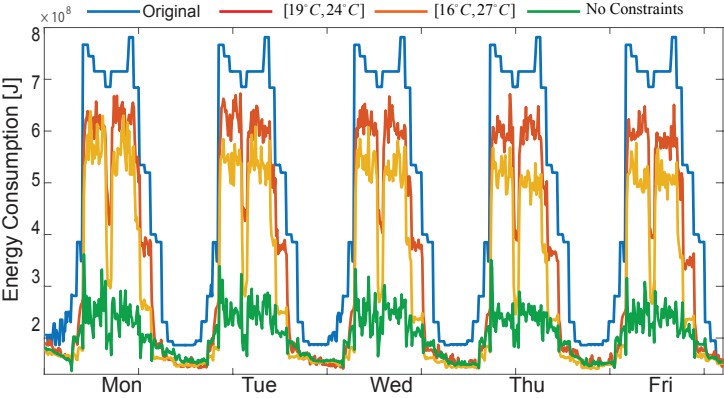

Figure 11: Results on one-week electricity usage of building using input convex neural network control method based upon different control constrains.

