# OpenReview forum: "Optimal Control Via Neural Networks: A Convex Approach"
_ICLR.cc/2019/Conference_

### Official Review · AnonReviewer2 · 2018-11-02
**A great work in quality, originality, significance, some questions to authors**

**Rating:** 7
**Confidence:** 4

**Review:**

The paper proposes neural networks which are convex on inputs data to control problems. These types of networks, constructed based on either MLP or RNN, are shown to have similar representation power as their non-convex versions, thus are potentially able to better capture the dynamics behind complex systems compared with linear models. On the other hand, convexity on inputs brings much convenience to the later optimization part, because there are no worries on global/local minimum or escaping saddle points. In other words, convex but nonlinear provides not only enough search space, but also fast and tractable optimization. The compromise here is the size of memory, since 1) more weights and biases are needed to connect inputs and hidden layers in such nets and 2) we need to store also the negative parts on a portion of weights.

Even though the idea of convex networks were not new, this work is novel in extending input convex RNN and applying it into dynamic control problems. As the main theoretical contribution, Theorem 2 shows that to have same representation power, input convex nets use polynomial number of activation functions, compared with exponential from using a set of affine functions. Experiments also show such effectiveness. The paper is clearly and nicely written. These are reasons I suggest accept.


Questions and suggestions:

1) For Lemma 1 and Theorem 1, I wonder whether similar results can be established for non-convex functions. Intuitively, it seems that as long as assuming Lipschiz continuous, we can always approximate a function by a maximum of many affine functions, no matter it is convex or not. Is this right or something is missing?

2) In the main paper, all experiments were aimed to address ICNN and ICRNN have good accuracy, but not they are easier to optimize due to convexity. In the abstract, it is mentioned "... using 5X less time", but I can only see this through appendix. A suggestion is at least describing some results on the comparison with training time in the main paper.

3) In Appendix A, it seems the NN is not trained very well as shown in the left figure. Is this because the number of parameters of NN is restricted to be the same as in ICNN? Do training on both spend the same resource, ie, number of epoch? Such descriptions are necessary here.

4) In Table 2 in appendix, why the running time of ICNN increases by a magnitude for large H in Ant case?


Typos:
	Page 1 "simple control algorithms HAS ..."
	Page 7 paragraph "Baselines": "Such (a) method".
	In the last line of Table 2, 979.73 should be bold instead of 5577.
	There is a ?? in appendix D.4.

---

> ### Author Response · Authors · 2018-11-18
> **Response to Reviewer 2**
>
> We are grateful to the reviewer for thoroughly reading our paper and providing these encouraging
> words. Below, we respond to the comments in detail.
>
> 1) For Lemma 1 and Theorem 1, I wonder whether similar results can be established for non-convex functions. Intuitively, it seems that as long as assuming Lipschiz continuous, we can always approximate a function by a maximum of many affine functions, no matter it is convex or not. Is this right or something is missing?
>
> This is an interesting and subtle question. If we restrict ourselves to “maximum” of affine functions, then we cannot construct functions that are not convex. This is from the fact that a pointwise max of convex functions (which include affine functions) is convex. As the reviewer points out, if we allow other types of operations, we can construct other types of functions. For example, if we change the pointwise max to the pointwise min, then we can approximate all Lipschiz concave functions. If we allow both max and min, we get all Lipschiz functions, but this just recover the result that neural networks can approximate most function types. We anticipate that different applications may require different function types to be approximated, and this is an active research direction for us.
>
> 2) In the main paper, all experiments were aimed to address ICNN and ICRNN have good accuracy, but not they are easier to optimize due to convexity. In the abstract, it is mentioned "... using 5X less time", but I can only see this through appendix. A suggestion is at least describing some results on the comparison with training time in the main paper.
>
> We thank the reviewer for pointing out this piece of missing information on running time in the main text. In the revised manuscript, we have added discussions on computation time in Section 4.1 to show our controller design would achieve both computation efficiency and performance improvement.
>
> 3) In Appendix A, it seems the NN is not trained very well as shown in the left figure. Is this because the number of parameters of NN is restricted to be the same as in ICNN? Do training on both spend the same resource, ie, number of epoch? Such descriptions are necessary here.
>
> In the toy example on classifying points in 2D grid, we used a 2-layer neural networks for both conventional neural networks and ICNN, with 200 neurons in each layer. We simulate the case when training data is small (100 training samples). We observe the results given by conventional neural networks are quite unstable by using different random seeds and are prone to be overfitting. On the contrary, by adding constraints on model weights to train the ICNN, fitting result is better using this small-size training data, while the learned landscape is also beneficial to the optimization problem.
> In the revised manuscript, we added more details on the model and training setup, the learning task, and the optimization task to address the confusion.
>
> 4) In Table 2 in appendix, why the running time of ICNN increases by a magnitude for large H in Ant case?
>
> We apologize for the typo in the case of Ant for computation time and the confusion it caused. We wanted to report everything in minutes but forgot to convert the time for the Ant case from seconds to minutes. In the revised version we have unified the running time under minutes.
>
> We also thank the reviewer for carefully proofreading the paper and extracting the typos.

---

### Official Review · AnonReviewer3 · 2018-11-02
**A good paper that bridges the gap between neural networks and MPC.**

**Rating:** 8
**Confidence:** 4

**Review:**

This paper proposes to use input convex neural networks (ICNN) to capture a complex relationship between control inputs and system dynamics, and then use trained ICNN to form a model predictive control (MPC) problem for control tasks.
The paper is well-written and bridges the gap between neural networks and MPC.
The main contribution of this paper is to use ICNN for learning system dynamics. ICNN is a neural network that only contains non-negative weights. Thanks to this constraint, ICNN is convex with respect to an input, therefore MPC problem with an ICNN model and additional convex constraints on control inputs is a convex optimization problem.
While it is not easy to solve such a convex problem, it has a global optimum, and a gradient descent algorithm will eventually reach such a point. It should also be noted that a convex problem has a robustness with respect to an initial starting point and an ICNN model itself as well. The latter is pretty important, since training ICNN (or NN) is a non-convex optimization, so the parameters in trained ICNN (or NN) model can vary depending on the initial random weights and learning rates, etc. Since a convex MPC has some robustness (or margin) over an error or deviation in system dynamics, while non-convex MPC does not, using ICNN can also stabilize the control inputs in MPC.
Overall, I believe that using ICNN to from convex MPC is a sample-efficient, non-intrusive way of constructing a controller with unknown dynamics. Below are some minor suggestions to improve this paper.

-- Page 18, there is Fig.??. Please fix this.
-- In experiments, could you compare the result with a conventional end-to-end RL approach? I know this is not a main point of this paper, but it can be more compelling.

---

> ### Author Response · Authors · 2018-11-18
> **Response to Reviewer 3**
>
> We thank the reviewer for the encouraging words, and we are also expecting our work would be able to serve as an efficient framework for incorporating deep learning into real-world control problems. The reviewer’s comments on the robustness of proposed convex MPC are also quite valuable. We would try to explore in details about the learning errors and control robustness in the future work.
>
> Here are some responses to the reviewer’s comments:
>
> -The miss-replacement of Figure in Page 18
> We thank the reviewer for pointing this out. In the revised version, we have added the fitting result comparison (Fig. 7 in Appendix D.4) for ICNN and a normal neural network, which shows that ICNN is able to learn the MuJoCo dynamics efficiently.
>
> -The comparison with end-to-end RL approach
>
> We thank the reviewer for this helpful suggestion. Conventional end-to-end RL approach directly learns the mapping from observations to actions without learning a system dynamics model. Such algorithms could achieve better performances, but are at the expense of much higher sample complexity. The model-free approach we compare with is the rllab implementation of trust region policy optimization (TRPO) [JL], which has obtained state-of-the-art results on MuJuCo tasks. We added Fig. 9 in Appendix D of the revised paper to compare our results with TRPO method and random shooting method [Nagabandi et al]. TRPO suffers from very high sample complexity and often requires millions of samples to achieve good performance. But here we only provided very few rollouts (since for physical system control, the sample collection might be limited by real-time operations, or it is difficult to explore the whole design space because suboptimal actions would lead to disastrous results), therefore, the performance by ICNN is much better than TRPO. Similarly to the model-based, model-free (MBMF) approach mentioned in [Nagabandi et al], the learned controller via ICNN could also provide good rollout samples and serve as a good initialization point for model-free RL method.
>
>
> References
> [JL] Schulman, John, Sergey Levine, Pieter Abbeel, Michael Jordan, and Philipp Moritz. "Trust region policy optimization." In International Conference on Machine Learning, pp. 1889-1897. 2015.

---

### Official Review · AnonReviewer1 · 2018-11-03
**Well-motived but may have serious issues. EDIT: Serious issues have been fixed.**

**Rating:** 6
**Confidence:** 3

**Review:**

This is a well-motived paper that considers bridging the gap
in discrete-time continuous-state/action optimal control
by approximating the system dynamics with a convex model class.
The convex model class has more representational power than
linear model classes while likely being more tractable and
stable than non-convex model classes.
They show empirical results in Mujoco continuous-control
environments and in an HVAC example.

I think this setup is a promising direction but I have
significant concerns with some of the details and claims
in this work:

1. Proposition 2 is wrong and the proposed input-convex recurrent
   neural network architecture not input-convex.
   To fix this, the D1 parameters should also be non-negative.
   To show why the proposition is wrong, consider the convexity of y2
   with respect to x1, using g to denote the activation function:

       z1 = g(U x1 + ...)
       y2 = g(D1 z1 + ...)

   Thus making

       y2 = g(D1 g(U x1 + ...) + ...)

   y2 is *not* necessarily convex with respect to x1 because D1 takes
   an unrestricted weighted sum of the convex functions g(U x1 + ...)

   With the ICRNN architecture as described in the paper not being
   input-convex, I do not know how to interpret the empirical findings
   in Section 4.2 that use this architecture.

2. I think a stronger and more formal argument should be used to show
   that Equation (5) is a convex optimization problem as claimed.
   It has arbitrary convex functions on the equality constraints that
   are composed with each other and then used in the objective.
   Even with parts of the objective being convex and non-decreasing
   as the text mentions, it's not clear that this is sufficient when
   combined with the composed functions in the constraints.

3. I have similar concerns with the convexity of Equation (6).
   Consider the convexity of x3 with respect to u1, where g is
   now an input-convex neural network (that is not recurrent):

       x3 = g(g(x1, u1), u2)

   This composes two convex functions that do *not* have non-decreasing
   properties and therefore introduces an equality constraint that
   is not necessarily even convex, almost certainly making the domain
   of this problem non-convex. I think a similar argument can be
   used to show why Equation (5) is not convex.

In addition to these significant concerns, I have a few other
minor comments.

1. Figure 1 hides too much information. It would be useful to know,
   for example, that the ICNN portion at the bottom right
   is solving a control optimization problem with an ICNN as
   part of the constraints.

2. The theoretical results in Section 3 seem slightly out-of-place within
   the broader context of this paper but are perhaps of standalone interest.
   Due to my concerns above I did not go into the details in this portion.

3. I think more information should be added to the last paragraph of
   Section 1 as it's claimed that the representational power of
   ICNNs and "a nice mathematical property" help improve the
   computational time of the method, but it's not clear why
   this is and this connection is not made anywhere else in the paper.

4. What method are you using to solve the control problems in
   Eq (5) and (6)?

5. The empirical setup and tasks seems identical to [Nagabandi et al.].
   Figure 3 directly compares to the K=100 case of their method.
   Why does Fig 6 of [Nagabandi et al.] have significantly higher rewards
   for their method, even in the K=5 case?

6. In Figure 5, f_NN seems surprisingly bad in the red region of the
   data on the left side. Is this because the model is not using
   many parameters? What are the sizes of the networks used?

---

> ### Author Response · Authors · 2018-11-18
> **Response to Reviewer 1-Theoretical guarantees on the convexity of neural networks-MPC [1/3]**
>
> We are grateful to the reviewer for carefully reading our paper and providing many helpful suggestions and comments that have significantly improved the revised version. We also appreciate the opportunity to clarify our presentation of theorems, figures and experiment setups, as well as some unclear writing in the manuscript. We agree with the reviewer that the original manuscript contained several parts that were not clear and some typos, and it resulted in some confusion on the technical results. Overall, we note that the results of the paper remain unchanged: deep (recurrent) neural networks can be made input convex and effectively used in control of complex systems. Based on the comments made by the reviewers, we have made the figures more illustrative, and the formulations and the theorems more rigorous. Our implementation of the algorithms is consistent with the updated manuscript, so we stress that these changes are made to clarify the writing of the paper and all of the simulation and numerical results remain unchanged. Below we provide a point-by-point account of the comments.
>
> -	Concerns on the correctness of Proposition 2
>
> We thank the reviewer for bringing up this important question and agree this was a point of confusion in our original manuscript. In Proposition 2 of the original submission, we stated that we only need to keep V and W non-negative and this will result in a network that is convex from input to output. This is true for a single step, but as the reviewer correctly points out, negative weights cannot go through composition and maintain convexity. Actually, Proposition 1 and Proposition 2 in our original submission give the sufficient condition for a network to be input-convex for a single step; when used for control purpose, these network structures (both ICNN and ICRNN) should be modified to their equivalent variants: restricting all weight matrices to be non-negative (element-wise) and augmenting the input to include its negation. Such network structure variants and “duplicate trick” have been mentioned in Section 3.1 Sketch of proof for Theorem 1 in our original manuscript, “We first construct a neural network with ReLU activation functions and both positive and negative weights, then we show that the weights between different layers of the network can be restricted to be nonnegative by a simple duplication trick. Specifically, since the weights in the input layer and passthrough layers can be negative, we simply add a negation of each input variable (e.g. both x and −x are given as inputs) to the network”. We apologize for not making this point clear and the notational confusions in our previous manuscript. To clarify, for both the MuJoCo locomotion tasks and the building control experiments, we used the modified input-convex network structures with all weights non-negative and input negation duplicates instead of the conventional input-convex structure for single step (but these two structures could be equivalently transformed).
> In the revised paper, we explicitly explain the sufficient conditions for ICNN/ICRNN variants that can be used for control purpose. We also update Proposition 1 and 2 to ease the confusions of convexity under control settings. Also, we have updated Figure 2 accordingly to demonstrate the modified ICNN/ICRNN structure, input duplication, operations and activation functions used for our control settings. For all the empirical experiments, we will release our code after the openreview process for result validation, which demonstrated that proposed control framework via input-convex networks obtain both good identification accuracies and better control performance compared with regular neural networks or linear models.

---

> > ### Author Response · Authors · 2018-11-18
> > **Theoretical guarantees on the convexity of neural networks-MPC [2/3]**
> >
> > -	A stronger and more formal argument should be used to show that Equation (5) is a convex optimization problem as claimed.
> >
> > We thank the reviewer for this helpful suggestion and agree that a more rigorous argument should be used to show that Equation (5) is a convex optimization problem. In the revised manuscript, we update Equation (5) to reflect the fact that we are using input-convex neural networks with all non-negative weights and the input negation trick. Equations (5d) and (5e) are added in the revised formulation which denote the augmented input variables and the consistency condition between u and its negation v.
> >
> > Then, in order to show Equation (5) is a convex optimization problem, we need to both the objective function and constraints are convex. Specifically,
> > (i). The objective function J(\hat{x},y) (Equation(5a)) is convex, non-decreasing with respect to \hat{x} and y;
> > (ii). The functions f and g are parameterized as ICRNNs with all weight matrices non-negative, which ensures f and g are convex and non-decreasing. Therefore rolling it out over time, the compositions remain convex with respect to the input.
> > (iii). The consistency constraint (5e) that one variable is the negation of the other is linear, therefore it preserves the convexity of optimization problems.
> >
> > We have clarified this discussion in the revised manuscript.
> >
> > -	Convexity on Equation (6)
> >
> > As a similar case to the optimization problem in Equation (5),  the system dynamics is governed  by Equation (6b). By restricting all weight matrices in ICNN to be non-negative and expanding  the inputs, the MPC formulation for MuJuCo case is convex with respect to control action vectors at different time. As shown in Fig. 3,  such convex properties also guaranteed that our  results on a series of control tasks outperformed current neural network based dynamical model.
> >
> >
> > Response to reviewer’s other comments:
> > -	Figure 1 hides too much information
> > We agree and have revised Figure 1 to include more information about problem setup related to modeling objective, control objective and constraints. In the left plot of revised Figure 1, we describe how an input convex neural network can be trained to learn the system dynamics. Then the right plot demonstrates the overall control framework, where we solve a convex predictive control problem to find the optimal actions. The optimization steps are also based on objectives and dynamics constraints represented by the trained networks.
> >
> > -	The theoretical results in Section 3 seem slightly out-of-place within the broader context of this paper
> >
> > We thank the reviewer for this question. The key idea for this section is by making the neural network convex from input to output, we are able to obtain both good predictive accuracies and tractable computational optimization problems. There are two main results presented in Section 3, Theorem 1 is about the representation capacity of ICNN (can represent all convex functions) and Theorem 2 is on the representation efficiency of ICNN (can be exponentially more efficient than conventional convex piecewise linear fitting [MP]).
> >
> > Since our proposed control framework involves two stages: 1) using ICNN/ICRNN for system identification; 2) design an optimal controller via solving a predictive control problem. For the system identification stage, obviously, one benefit of using input convex networks (instead of conventional neural networks) is its computational trackability and optimality guarantee for the subsequent optimization stage. However, besides the trackability, reasonable representation capacity to model complex relationships is also desired as a system identification model. Theorem 1 and 2, on this aspect, demonstrate such representability and efficiency of ICNN. In the revised manuscript, we have added the above discussion at the beginning of Section 3 to improve the coherence of the paper.

---

> > > ### Author Response · Authors · 2018-11-18
> > > **Theoretical guarantees on the convexity of neural networks-MPC [3/3]**
> > >
> > > -      More information should be added to the last paragraph of Section 1 as it's claimed that the representational power of ICNNs and "a nice mathematical property" .
> > >
> > > We have added the following discussion to the end of Section 1: Our method enjoys computational efficiency in two perspectives. Firstly, as stated in Theorem 2, compared to model based method which often employs piecewise linear functions, we could train ICNN or ICRNN (with exponentially less variables/pieces) using off-the-shelf deep learning packages such as PyTorch and Tensorflow, while the optimal control can be achieved by solving convex optimization; Secondly, compared to model-free (deep) reinforcement learning algorithms, which usually takes an end-to-end settings and requires lots of samples and long training time, our model is learning and controlling based on the system dynamics – this can be much more sample efficient. There is also an ongoing debate on the model-free and model-based reinforcement learning algorithms [BR], and we look forward to incorporating learning into control tasks with optimality guarantees.
> > >
> > > -	What method are you using to solve the control problems in Eq (5) and (6)?
> > >
> > > In Eq (5) and (6), since both the objectives and the constraints contain neural networks, we set up our networks with Tensorflow and solve the control problem using projected gradient descent method with adaptive step size. The gradients can be calculated via existing modules in Tensorflow for backpropagation. In both cases the optimization problems can be solved fairly fast and we observe the solution convergence after around 20 iterations. As shown in Figure 4 (c) of the original manuscript, the control actions outputted by solving (5) are stable and much better than the results achieved from regular neural network + MPC (which has no optimality guarantee). In the revised paper, we have included more details on the solution algorithm of Eq. (5) in the last paragraph of Section 2.
> > >
> > > -	Why does Figure 6 of [Nagabandi et al.] have significantly higher rewards for their method, even in the K=5 case?
> > >
> > > We thank the reviewer for carefully proofreading the figure, and we also thank Nagabandi et al open sourced their code. We re-run their simulations using all the default parameters and observed the reward for their cases are all around 10x less than they showed in Figure 6. We also refer to [KC], where their rewards on swimmer case are significantly smaller than the case as [Nagabandi et al]. We are not sure what is causing the difference in the performances, although we believe there may be difference in hyperparameter settings and the random starting points between our and [Nagabandi et al]’s result. We make sure the comparison in Figure 3 of our paper are using the same hyperparameters and training data, except the differences on control methods.
> > >
> > > -	The experimental results in Figure 5
> > >
> > > We thank the reviewer for bringing up this interesting point. In the toy example on classifying points in 2D grid, we used a 2-layer neural networks for both conventional neural networks and ICNN, with 200 neurons in each layer. We simulate the case when training data is small (100 training samples). We observe the results given by conventional neural networks are quite unstable by using different random seeds and are prone to be overfitting. On the contrary, by adding constraints on model weights to train the ICNN, fitting result is better using this small-size training data, while the learned landscape is also beneficial to the optimization problem.
> > > In the revised manuscript, we added more details on the model and training setup, the learning task, and the optimization task to address the confusion.
> > >
> > > References
> > >
> > > [MP] Alessandro Magnani and Stephen P Boyd. “Convex piecewise-linear fitting”. Optimization and Engineering, 10(1):1–17, 2009.
> > >
> > > [BR] Recht, Benjamin. "A tour of reinforcement learning: The view from continuous control." arXiv preprint arXiv:1806.09460(2018).
> > >
> > > [KC] Kurutach T, Clavera I, Duan Y, Tamar A, Abbeel P. “Model-Ensemble Trust-Region Policy Optimization”. arXiv preprint arXiv:1802.10592. 2018 Feb 28.

---

> > > > ### Comment · AnonReviewer1 · 2018-12-05
> > > > **Excellent revised paper.**
> > > >
> > > > Thanks for the thorough response and revised version of the paper.
> > > > The updates are commendable and I apologize for the delays from my end
> > > > as I needed the time to thoroughly look over the new manuscript.
> > > > I have updated my score from a 1 to a 6.
> > > > The revised paper no longer has the significant errors with convexity
> > > > that I found in my original review and I think that the models,
> > > > experimental tasks, and analysis provide a useful contribution
> > > > to the community.
> > > >
> > > > One limitation that is now present in the revised version of the
> > > > paper that was not present in the original submission is that
> > > > these dynamics models do *not* subsume linear dynamics models.
> > > > This is because the dynamics are being approximated with
> > > > convex and non-decreasing functions over the state space,
> > > > while linear models *are* able to model decreasing functions
> > > > over the state space while retaining convexity of the overall
> > > > control problem. I would like an updated version of this paper
> > > > to highlight this limitation of the method as I expect it to
> > > > hurt some applications (although it is fine in other contexts.)
> > > >
> > > > On the presentation of the work, I think it should be made clearer
> > > > that the motivation of the "input duplication trick" over the
> > > > control space is to restrict the networks to be non-decreasing
> > > > over the state space while not restricting the control space.
> > > >
> > > > I think the duplicated inputs unnecessarily complicates the
> > > > presentation at parts, such as the borderline-misleading
> > > > statement at the end of Section 2.1 that says:
> > > >
> > > >    Note that such construction guarantees that the
> > > >    network is convex and non-decreasing with respect
> > > >    to the expanded inputs \hat u = [u -u]
> > > >
> > > > This part is almost misleading because the model is
> > > > *not* non-decreasing with respect to the controls u,
> > > >
> > > > I have a minor concern/question on the locomotion experiment:
> > > > If I understand it correctly, there are two differences to
> > > > Nagabandi et al.:
> > > >
> > > > 1) The dynamics are input-convex, and
> > > > 2) The inference procedure uses gradient descent over
> > > >    the action space of the control problem that
> > > >    Nagabandi does with just random search
> > > >
> > > > It's not clear which one of these is improving the accuracy,
> > > > as taking a few gradient steps over the non-convex MPC problem
> > > > in Nagabandi et al. is also reasonable and would likely also
> > > > improve the performance, even if the true optimum is not reached.
> > > > Did you try comparing to this as a baseline?
> > > >
> > > > On the energy consumption experiment, is the RC model a
> > > > linear dynamics model that only looks at single-step
> > > > states and actions as g(x_t, u_t)?
> > > > If so, comparing an ICNN that uses a previous trajectory
> > > > as g(x_{t-n_w:t}, u_{t-n_w,t}) to this seems somewhat
> > > > unfair as another reasonable baseline would be a linear
> > > > model that also uses the previous trajectory.
> > > >
> > > > It could be interesting to include some portions in the paper
> > > > about the ways you've seen the non-convex MPC with the RNN
> > > > dynamics model fail that the ICNN model overcomes.
> > > > For example, does the lack of smoothness cause the control problem to
> > > > get stuck in bad local minimum?
> > > >
> > > > I still see section 3 as being very out-of-place within the
> > > > broader context of this paper and I have not reviewed this
> > > > portion of the paper.
> > > >
> > > > As a minor comment, please add parenthetical citations where
> > > > appropriate to the paper.

---

> > > > > ### Author Response · Authors · 2018-12-09
> > > > > **Response to Reviewer 1- Great insights and suggestions for our future work [1/2]**
> > > > >
> > > > > We thank the reviewer again for carefully reading our manuscript and providing so many valuable feedbacks. We address reviewer’s concern as follows:
> > > > >
> > > > > 1) One limitation that is now present in the revised version of the paper that was not present in the original submission is that these dynamics models do *not* subsume linear dynamics models. This is because the dynamics are being approximated with convex and non-decreasing functions over the state space, while linear models *are* able to model decreasing functions over the state space while retaining convexity of the overall control problem. I would like an updated version of this paper to highlight this limitation of the method as I expect it to hurt some applications (although it is fine in other contexts).
> > > > >
> > > > > We thank the reviewer for this insightful comment and we agree that the proposed input convex neural networks do not subsume linear dynamics models completely. Specifically, the proposed ICNN/ICRNN could only capture the dynamics convex and non-decreasing over the state space. But since we are not restricting the control space (system state at any time can be viewed as a function of the initial system state and all previous control inputs if one unrolls the system dynamics equation entirely), and we have explicitly included multiple previous states in the state transition dynamics $s_t = g(s_{t-n_w:t-1}, u_{t-n_w:t})$, so the non-decreasing constraint should not hurt the representation capacity by much.
> > > > >
> > > > > In the revised manuscript, we add the following discussion in page 5 under Eq. (5) to emphasize the differences between input convex neural networks and linear models. “Note that as a general formulation, we do not include the duplication tricks on state variables, so the dynamics fitted by Eq. (5b) and (5c) are non-decreasing over state space, which are not equivalent to those dynamics represented by linear systems. However, since we are not restricting the control space (dynamics can be both increasing or decreasing on control variables), and we have explicitly included multiple previous states in the system transition dynamics, so the non-decreasing constraint over state space should not restrict the representation capacity by much. In Section 3, we theoretically prove the representability of proposed networks.”
> > > > >
> > > > > 2) I think it should be made clearer that the motivation of the ‘input duplication trick’ over the control space is to restrict the networks to be non-decreasing over the state space while not restricting the control space. I think the duplicated inputs unnecessarily complicates the presentation at parts e.g.  the end of Section 2.1.
> > > > >
> > > > > We thank the reviewer for pointing out this misleading presentation at the end of Section 2.1 about the duplication. In the revised manuscript, we write out explicitly the convex and non-decreasing properties over the expanded control variable \hat{u} rather than u.

---

> > > > > > ### Author Response · Authors · 2018-12-09
> > > > > > **Great insights and suggestions for our future work [2/2]**
> > > > > >
> > > > > > 3)  I have a minor concern/question on the locomotion experiment: If I understand it correctly, there are two differences to Nagabandi et al.: the dynamics are input-convex, and the inference procedure uses gradient descent over the action space of the control problem that Nagabandi does with just random search. It's not clear which one of these is improving the accuracy, as taking a few gradient steps over the non-convex MPC problem in Nagabandi et al. is also reasonable and would likely also improve the performance, even if the true optimum is not reached. Did you try comparing to this as a baseline?
> > > > > >
> > > > > > We agree with the reviewer’s insights that both convexity and gradient descent steps could bring benefits in the locomotion tasks. Similar to the energy consumption case, we also tried to directly do gradient steps using a normal neural network to represent dynamics. It would improve the reward than the random search method but the reward is lower than our proposed ICNN method. We are still working on more experiments to gain a deeper understanding of the contributions from these two factors, and the general tradeoff between model accuracy and solution traceability in model predictive control (MPC) would be an important direction for our future work. Moreover, since of computation efforts of Nagabandi et al’s method have been taken on the random search step, to make a fair comparison in the paper we only show the comparison with their original method on the task rewards and computation time.
> > > > > >
> > > > > > 4) On the energy consumption experiment, is the RC model a linear dynamics model that only looks at single-step states and actions as g(x_t, u_t)? If so, comparing an ICNN that uses a previous trajectory as g(x_{t-n_w:t}, u_{t-n_w,t}) to this seems somewhat unfair as another reasonable baseline would be a linear model that also uses the previous trajectory. It could be interesting to include some portions in the paper about the ways you've seen the non-convex MPC with the RNN dynamics model fail that the ICNN model overcomes. For example, does the lack of smoothness cause the control problem to get stuck in bad local minimum?
> > > > > >
> > > > > > The RC model is linear dynamics using single-step states, yet has been used as a standard method in building energy management. In order to demonstrate where the non-convex MPC with the RNN dynamics model fails and the ICNN model overcomes, we actually included a comparison of the control performance of ICRNN with normal RNN in Figure. 4. We think the most interesting result is shown in Fig. 4(c). By using ICRNN, the final control actions (in red) is stable, while control signals founds by normal RNN (in green) have many oscillations, which seems to be stuck in some local minima, and such drastic control variations is not desirable for physical system control.
> > > > > >
> > > > > > 5) As a minor comment, please add parenthetical citations where appropriate to the paper.
> > > > > >
> > > > > > We thank the reviewer for this helpful suggestion and we have modified the citation format in the revised paper.

---

### Public Comment · (anonymous) · 2018-09-28
**Efficient model based control of almost convex systems -- convexity comes in surprisingly**

This paper proposed a powerful tool for an important category of model based control system. For model based control, there are usually two steps: 1. modeling the system as accurate as you can, and 2. Optimize over the fitted system to find the best control strategy. It is known that convex optimization is always tractable, so if the true system can be convex, or almost exactly regressed by a convex function, we can make use of it. However, for modeling complex systems, where neural network becomes more popular, there is no guarantee that step 1 outputs a convex system -- even if the system is convex, we do not know whether the model is convex unless we can prove that they are close enough, which is usually difficult. So paper such as https://arxiv.org/abs/1708.02596 use a non-convex model and have to search for the control strategy by griding and testing the entire space almost exhaustedly.

This paper propose an NN structure which is simply based on only Relu, but guarantees a convex modeling of the system. Compared with maximum piecewise linear modeling, it only introduces a fixed 2 piece linear activating module, but dramatically decreases the number of parameters from exponential to polynomial, which makes it realizable.

But if it's true, now something interesting might come up. The authors show that even for seemingly hard scenarios including Mujoco, it performances well to attempt a convex model, whose optimizer corresponds to good control scheme in true problems. It likely means that, despite the impossibility to show the model is everywhere accurate, the model well sketches the landscape of true loss around its minimizer (where is probably locally convex) and the trajectory of optimizing iterations, where people are interested. I'm not sure if that's true. NN always surprise people, but it's definitely worth rethinking and experimenting, based on the result on such complex tasks in this paper.

---

### Author Response · Authors · 2018-11-18
**Revised Paper Uploaded to Address Reviewers' Feedback**

We would like to thank all the reviewers for their constructive comments. We have responded to each reviewer’s comments individually, and in summary, we have made the following clarifications or changes:
-Clarify the inputs and constraints on the input convex neural network weights
-Update texts, equations accordingly to avoid notation confusions
-Explain and add details for simulation results.

---

### Comment · Area_Chair1 · 2018-11-20
**reviewers: comments on responses and revised-paper improvements?**

The detailed reviews and responses are commendable.  Thanks to all.

Reviewers:  can you comment on whether the revised-paper and author responses have addressed  your concerns?
In particular, for reviewer 1, this would be important. Note that the revised version can also be viewed in a way that lets one easily see the differences.

-- area chair

---

### Public Comment · (anonymous) · 2019-10-08
**Source code available?**

Is the source code available?

---

### Meta-Review · Area_Chair1 · 2018-12-15
**strong paper; nomination for oral presentation; nomination for best reviewer**

**Confidence:** 4
**Recommendation:** Accept (Poster)

**Metareview:**

The paper makes progress on a problem that is still largely unexplored, presents promising results, and builds bridges with
prior work on optimal control.  It designs input convex recurrent neural networks to capture temporal behavior of
dynamical systems; this then allows optimal controllers to be computed by solving a convex model predictive control problem.

There were initial critiques regarding some of the claims. These have now been clarified.
Also, there is in the end a compromise between the (necessary) approximations of the input-convex model and the true dynamics, and being able to compute an optimal result.

Overall, all reviewers and the AC are in agreement to see this paper accepted.
There was extensive and productive interaction between the reviewers and authors.
It makes contributions that will be of interest to many, and builds interesting bridges with known control methods.